# Improving Uncertainty Estimation through Semantically Diverse Language Generation

**Lukas Aichberger**[1], **Kajetan Schweighofer**[1], **Mykyta Ielanskyi**[1], **Sepp Hochreiter**[1,2]

[1] ELLIS Unit Linz and LIT AI Lab, Institute for Machine Learning,
Johannes Kepler University Linz, Austria

[2] NXAI GmbH, Linz, Austria

`{aichberger, schweighofer, ielanskyi, hochreit}@ml.jku.at`

## Abstract

Large language models (LLMs) can suffer from hallucinations when generating text. These hallucinations impede various applications in society and industry by making LLMs untrustworthy. Current LLMs generate text in an autoregressive fashion by predicting and appending text tokens. When an LLM is uncertain about the semantic meaning of the next tokens to generate, it is likely to start hallucinating. Thus, it has been suggested that predictive uncertainty is one of the main causes of hallucinations. We introduce Semantically Diverse Language Generation (`SDLG`) to quantify predictive uncertainty in LLMs. `SDLG` steers the LLM to generate semantically diverse yet likely alternatives for an initially generated text. This approach provides a precise measure of aleatoric semantic uncertainty, detecting whether the initial text is likely to be hallucinated. Experiments on question-answering tasks demonstrate that `SDLG` consistently outperforms existing methods while being the most computationally efficient, setting a new standard for uncertainty estimation in LLMs.

## 1 Introduction

Hallucinations hinder a broad use of LLMs in practical applications and critical decision-making processes as they make them untrustworthy (Manakul et al., 2023). Hallucinations are fragments of generated text that, despite appearing cohesive, are not factual. At the time of writing, there is no consensus on the exact nature of all causes of hallucination. We consider generated text to be hallucinated if it stems from contradictory or non-existent facts in the training data or input prompt. Such hallucinations are conjectured to be mainly caused by the predictive uncertainty inherent to probabilistic models (Xiao and Wang, 2021). This type of hallucination is also referred to as confabulation (Farquhar et al., 2024), and we shall use the two terms interchangeably. While uncertainty estimation for classification tasks has been developed extensively (Hüllermeier and Waegeman, 2021; Gawlikowski et al., 2023), it remains a challenging problem for autoregressive tasks, particularly in natural language generation (NLG).

Uncertainty estimation in NLG involves assessing the uncertainty of an initially generated text (output sequence) for a given prompt (input sequence). Current methods typically assess this uncertainty by generating multiple output sequences, usually with a single given language model (Xiao and Wang, 2021; Malinin and Gales, 2021; Kuhn et al., 2023; Lin et al., 2023; Duan et al., 2023; Manakul et al., 2023). Importantly, Kuhn et al. (2023) propose to consider semantic clusters rather than individual output sequences by grouping them according to their semantic equivalence. The *semantic* uncertainty of the initial output sequence should only increase if a language model is likely to generate alternative output sequences that differ in semantics. Hence, semantic uncertainty involves estimating the probability that a language model generates an output sequence belonging to a specific semantic cluster. Empirical results confirm that incorporating semantics into uncertainty estimation in language models achieves state-of-the-art performance (Kuhn et al., 2023). The current approach utilizes Monte Carlo (MC) approximation via multinomial sampling to generate alternative output sequences that are classified into semantic clusters by a natural language inference (NLI) model (Kuhn et al., 2023). Sample sizes typically range from the low double-digit range (Malinin and Gales, 2021; Kuhn et al., 2023; Duan et al., 2023) to only a few hundred for studies conducted

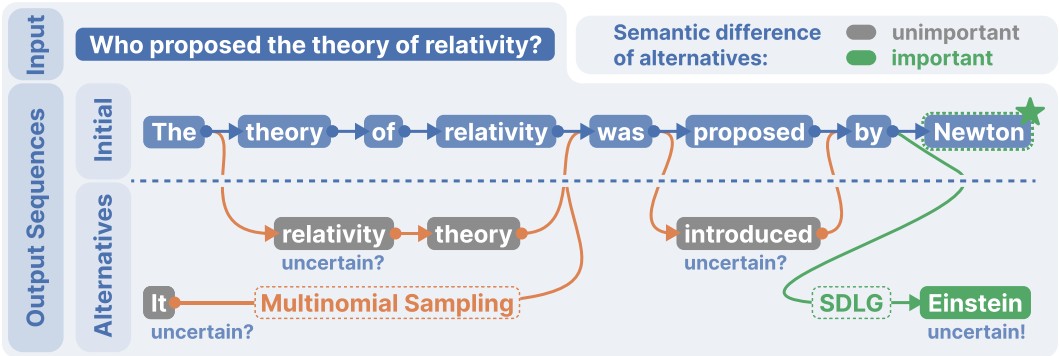

Figure 1: Using standard multinomial sampling to generate text does not account for its semantics. Thus, it relies on chance to obtain semantically diverse output sequences and is prone to miss them. SDLG addresses this by specifically searching for likely, but semantically different output sequences. Thereby, the estimation of semantic uncertainty in language models is improved.

on large-scale compute (Kadavath et al., 2022), as autoregressive generations are computationally expensive. This is suboptimal, as the current sampling methods are imprecise with a small number of samples (Bishop, 2006) and computationally too expensive to ensure high precision with a large number of samples.

To improve upon the limitations of the current state-of-the-art approach, we introduce Semantically Diverse Language Generation (SDLG), a method that efficiently estimates semantic uncertainty by utilizing importance sampling to generate output sequences. We introduce a proposal distribution that samples semantically diverse output sequences (see Fig. 1). SDLG utilizes the NLI model not only for transforming the space of generated output sequences to semantic clusters but also for computing the contribution of every token to the final semantics. Subsequently, the semantically most relevant tokens are substituted to explicitly steer the generation toward semantically diverse yet likely alternative output sequences, while correcting for the increased sampling probability using importance sampling. This can be viewed as stress-testing the language model, unveiling output sequences that are valuable summands for estimators of semantic uncertainty. They are generated by the current sampling methods only by chance, making SDLG a more systematic and reliable method for capturing semantic uncertainty. Our main contributions are:

- We propose a novel method for generating semantically diverse yet likely output sequences. Empirical results demonstrate that our method outperforms existing methods for uncertainty estimation in NLG, specifically across a variety of free-form question answering tasks.

- We establish a theoretical foundation for uncertainty measures in NLG and introduce theoretically grounded estimators for aleatoric semantic uncertainty, also known as semantic entropy. Applying these estimators enhances empirical performance of uncertainty estimation in language models.

## 2  MEASURING PREDICTIVE UNCERTAINTY IN NLG

Uncertainty estimation in classification tasks has already been well-established (Gal and Ghahramani, 2016). However, these measures cannot be directly applied to uncertainty estimation in NLG. Two key aspects, which differ from estimating uncertainty in classification tasks, have to be considered. First, a sequence of autoregressive predictions collectively forms the final output of a model. Second, unlike classification tasks where classes usually are mutually exclusive, different output sequences may be equivalent in their semantic meaning. To account for these aspects, Kuhn et al. (2023) introduce *semantic entropy*, yet without proper theoretical grounding. In the following, we derive semantic entropy starting from the well-studied uncertainty estimation in classification tasks.

**Predictive uncertainty in classification.** We briefly revisit uncertainty estimation for classification tasks. In this work, we quantify the predictive uncertainty of a single given "off-the-shelf" model. Given are a classification model parametrized by $w$ and an input vector $x$. The predictive distribution under the given model is denoted as $p(y \mid x, w)$. We assume a fixed dataset $\mathcal{D}$ that was sampled according to the predictive distribution $p(y \mid x, w^*)$ under true model parameters

$\boldsymbol{w}^*$. Thus, we assume that the model class can approximate the true distribution sufficiently well, a common and usually necessary assumption (Hüllermeier and Waegeman, 2021). The posterior distribution $p(\boldsymbol{w} \mid \mathcal{D})$ denotes how likely $\boldsymbol{w}$ matches $\boldsymbol{w}^*$. Following Schweighofer et al. (2023a;b), the predictive uncertainty of a single given model parametrized by $\boldsymbol{w}$ is given by

$$\underbrace{\mathrm{E}_{\tilde{\boldsymbol{w}}}\big[\mathrm{CE}(p(y \mid \boldsymbol{x}, \boldsymbol{w}); p(y \mid \boldsymbol{x}, \tilde{\boldsymbol{w}}))\big]}_{\text{total}} = \underbrace{\mathrm{H}(p(y \mid \boldsymbol{x}, \boldsymbol{w}))}_{\text{aleatoric}} + \underbrace{\mathrm{E}_{\tilde{\boldsymbol{w}}}\big[\mathrm{KL}(p(y \mid \boldsymbol{x}, \boldsymbol{w}) \,\|\, p(y \mid \boldsymbol{x}, \tilde{\boldsymbol{w}}))\big]}_{\text{epistemic}} \quad (1)$$

where $\mathrm{E}_{\tilde{\boldsymbol{w}}} = \mathrm{E}_{\tilde{\boldsymbol{w}} \sim p(\tilde{\boldsymbol{w}} \mid \mathcal{D})}$. The total uncertainty, given by the posterior expectation of the cross-entropy $\mathrm{CE}(\cdot; \cdot)$, is additively decomposed into aleatoric and epistemic uncertainty. The aleatoric uncertainty is the Shannon entropy $\mathrm{H}(\cdot)$ of the predictive distribution under the given model. The epistemic uncertainty is the posterior expectation of the Kullback-Leibler divergence $\mathrm{KL}(\cdot \| \cdot)$ between the given model and possible true models according to their posterior probability.

**Predictive uncertainty in NLG.** Given are an autoregressive language model parametrized by $\boldsymbol{w}$, a vocabulary $\mathcal{V}$, and an input sequence of tokens $\boldsymbol{x} = (x_1, ..., x_M)$ with $x \in \mathcal{V}$. An output of the language model is a sequence of tokens $\boldsymbol{y} = (y_1, ..., y_T) \in \mathcal{Y}$ with $y \in \mathcal{V}$. The predictive distribution at step $t$ of the output sequence $\boldsymbol{y}$ is conditioned on both the input sequence and all previously generated tokens, denoted as $p(y_t \mid \boldsymbol{x}, \boldsymbol{y}_{<t}, \boldsymbol{w})$. The probability of an output sequence is the product of the individual token probabilities: $p(\boldsymbol{y} \mid \boldsymbol{x}, \boldsymbol{w}) = \prod_{t=1}^{T} p(y_t \mid \boldsymbol{x}, \boldsymbol{y}_{<t}, \boldsymbol{w})$ (Sutskever et al., 2014). In practice, $p(\boldsymbol{y} \mid \boldsymbol{x}, \boldsymbol{w})$ is often length-normalized to not favor short output sequences (Cover and Thomas, 2006; Malinin and Gales, 2021), which results in $\bar{p}(\boldsymbol{y} \mid \boldsymbol{x}, \boldsymbol{w}) = \exp\left(\frac{1}{T} \sum_{t=1}^{T} \log p(y_t \mid \boldsymbol{x}, \boldsymbol{y}_{<t}, \boldsymbol{w})\right)$.

Evaluating the whole set of possible output sequences $\mathcal{Y}$ is usually intractable, as it scales exponentially with the sequence length $T$, thus $\mathcal{O}(|\mathcal{V}|^T)$. Furthermore, as mentioned previously, a language model that likely generates different output sequences from the same input sequence should not necessarily indicate high predictive uncertainty if the output sequences mean the same thing. Hence, predictive uncertainty should be considered high only when different output sequences also exhibit semantically diverse meanings (Kuhn et al., 2023). Instead of directly utilizing the distribution over output sequences $p(\boldsymbol{y} \mid \boldsymbol{x}, \boldsymbol{w})$, the distribution over semantic clusters

$$p(c \mid \boldsymbol{x}, \boldsymbol{w}) = \sum_{\mathcal{Y}} p(c \mid \boldsymbol{y}, \boldsymbol{x}, \boldsymbol{w}) \, p(\boldsymbol{y} \mid \boldsymbol{x}, \boldsymbol{w}) = \sum_{\mathcal{Y}} \mathbb{1}\{\boldsymbol{y} \in c \mid \boldsymbol{x}, \boldsymbol{w}\} \, p(\boldsymbol{y} \mid \boldsymbol{x}, \boldsymbol{w}) \quad (2)$$

is used to derive the predictive uncertainty in the NLG setting. Here, $p(\boldsymbol{y} \mid \boldsymbol{x}, \boldsymbol{w})$ expresses the probability of generating an output sequence $\boldsymbol{y}$ and $p(c \mid \boldsymbol{y}, \boldsymbol{x}, \boldsymbol{w})$ expresses the probability of $\boldsymbol{y}$ belonging to a certain semantic cluster $c \in \mathcal{C}$. Although a specific output sequence might potentially be attributed to more than one semantic cluster, we follow Kuhn et al. (2023); Farquhar et al. (2024) in assuming that each output sequence is attributed to a single semantic cluster. They demonstrate that assigning semantic clusters by predicting the semantic equivalence between generated output sequences with an NLI model empirically performs well. The NLI model takes two sequences as input and predicts whether they entail or contradict each other. Two output sequences are semantic equivalent if they entail each other in both orders and thus belong to the same semantic cluster by definition. Consequently, $p(c \mid \boldsymbol{y}, \boldsymbol{x}, \boldsymbol{w})$ is equivalent to $\mathbb{1}\{\boldsymbol{y} \in c \mid \boldsymbol{x}, \boldsymbol{w}\}$, where $\mathbb{1}\{\boldsymbol{y} \in c \mid \boldsymbol{x}, \boldsymbol{w}\} = 1$ iff $\boldsymbol{y}$ belongs to semantic cluster $c$. For the set of all possible output sequences $\mathcal{Y}$, $p(c \mid \boldsymbol{x}, \boldsymbol{w})$ expresses the probability of the language model generating an output sequence belonging to a specific semantic cluster for a given input sequence. Adopting the definition of predictive uncertainty in Eq. (1), the total predictive semantic uncertainty

$$\underbrace{\mathrm{E}_{\tilde{\boldsymbol{w}}}\big[\mathrm{CE}(p(c \mid \boldsymbol{x}, \boldsymbol{w}); p(c \mid \boldsymbol{x}, \tilde{\boldsymbol{w}}))\big]}_{\text{total}} = \underbrace{\mathrm{H}(p(c \mid \boldsymbol{x}, \boldsymbol{w}))}_{\text{aleatoric}} + \underbrace{\mathrm{E}_{\tilde{\boldsymbol{w}}}\big[\mathrm{KL}(p(c \mid \boldsymbol{x}, \boldsymbol{w}) \,\|\, p(c \mid \boldsymbol{x}, \tilde{\boldsymbol{w}}))\big]}_{\text{epistemic}} \quad (3)$$

can again be additively decomposed into aleatoric and epistemic semantic uncertainty. The epistemic semantic uncertainty is again a posterior expectation, which is particularly challenging to estimate for current language models with billions of parameters (Zhang et al., 2022; Touvron et al., 2023). The aleatoric semantic uncertainty turns out to be precisely the semantic entropy proposed by Kuhn et al. (2023). Semantic entropy is the entropy of the semantic cluster probability distribution,

$$\mathrm{H}(p(c \mid \boldsymbol{x}, \boldsymbol{w})) = -\sum_{\mathcal{C}} \log p(c \mid \boldsymbol{x}, \boldsymbol{w}) \, p(c \mid \boldsymbol{x}, \boldsymbol{w}) \quad (4)$$

under a single given language model. To determine if a language model is uncertain about its output, it is essential to accurately estimate its semantic entropy, as addressed in the following section.

## 3 ESTIMATING ALEATORIC SEMANTIC UNCERTAINTY IN NLG

We now discuss practical considerations regarding the estimation of the aleatoric semantic uncertainty, namely the semantic entropy as given in Eq. (4). First, we find that directly estimating it by obtaining samples from semantic clusters is theoretically not justified in the current setting. Instead, the underlying distribution of semantic clusters itself has to be estimated through sampling. Second, to effectively estimate this distribution of semantic clusters, we utilize importance sampling. These two insights contribute to a more accurate uncertainty estimation in NLG.

**Monte Carlo (MC) estimation.** Kuhn et al. (2023) propose to approximate the semantic entropy by directly using the MC estimator

$$\mathrm{H}(p(c \mid \boldsymbol{x}, \boldsymbol{w})) \;\approx\; -\frac{1}{N} \sum_{n=1}^{N} \log p(c^n \mid \boldsymbol{x}, \boldsymbol{w}), \quad c^n \sim p(c \mid \boldsymbol{x}, \boldsymbol{w}). \tag{5}$$

However, we cannot directly sample from $p(c \mid \boldsymbol{x}, \boldsymbol{w})$, but only from $p(\boldsymbol{y} \mid \boldsymbol{x}, \boldsymbol{w})$. Therefore, it is impossible to directly use the estimator in Eq. (5). Alternatively, one can first approximate the semantic cluster probability distribution

$$p(c \mid \boldsymbol{x}, \boldsymbol{w}) \;\approx\; \frac{1}{N} \sum_{n=1}^{N} \mathbb{1}\{\boldsymbol{y} \in c \mid \boldsymbol{x}, \boldsymbol{w}\}, \quad \boldsymbol{y}^n \sim p(\boldsymbol{y} \mid \boldsymbol{x}, \boldsymbol{w}). \tag{6}$$

Output sequences $\boldsymbol{y}^n$ are simply generated via multinomial sampling (Kuhn et al., 2023). This estimate of the semantic cluster probability distribution can directly be used to approximate the semantic entropy. Since usually not all clusters $c \in \mathcal{C}$ are found through sampling, the sum is taken over observed clusters $c_1, ..., c_M$ to which $\{\boldsymbol{y}^n\}_{n=1}^N$ were assigned by the NLI model, resulting in

$$\mathrm{H}(p(c \mid \boldsymbol{x}, \boldsymbol{w})) \;\approx\; -\sum_{m=1}^{M} \log p(c_m \mid \boldsymbol{x}, \boldsymbol{w}) \, p(c_m \mid \boldsymbol{x}, \boldsymbol{w}). \tag{7}$$

In Sec. 6, we empirically show that this proper estimator for semantic entropy outperforms the improper estimator implemented by Kuhn et al. (2023). For further details see Sec. C in the appendix.

**Importance sampling.** Due to the computational cost of autoregressively sampling from multi-billion-parameter language models, the sample size $N$ is usually kept very low in practice. However, this implies that the variance of the MC estimator in Eq. (6) remains high. To lower the variance, a standard technique is importance sampling according to a proposal distribution $q$ instead of the target distribution $p$ (Bishop, 2006). Eq. (6) thus changes to

$$p(c \mid \boldsymbol{x}, \boldsymbol{w}) \;\approx\; \frac{1}{N} \sum_{n=1}^{N} \mathbb{1}\{\boldsymbol{y} \in c \mid \boldsymbol{x}, \boldsymbol{w}\} \frac{p(\boldsymbol{y}^n \mid \boldsymbol{x}, \boldsymbol{w})}{q(\boldsymbol{y}^n \mid \boldsymbol{x}, \boldsymbol{w})}, \quad \boldsymbol{y}^n \sim q(\boldsymbol{y} \mid \boldsymbol{x}, \boldsymbol{w}). \tag{8}$$

The quality of the approximation strongly depends on the choice of the proposal distribution. It should closely approximate the target distribution and have good overlap, i.e. the proposal distribution should have probability mass everywhere the target distribution has substantial probability mass (Bishop, 2006). Thus, a good proposal distribution should cover semantic clusters with substantial probability mass under the target distribution. In the following section, we describe our method to construct an empirical proposal distribution $q(\boldsymbol{y} \mid \boldsymbol{x}, \boldsymbol{w})$ with high mass at important semantic clusters by explicitly promoting both the likelihood and diversity of output sequences.

## 4 SEMANTICALLY DIVERSE LANGUAGE GENERATION

Estimating the semantic entropy according to Eq. (7) requires approximating the semantic cluster probability distribution. This probability distribution can either be approximated according to Eq. (6), or via importance sampling according to Eq. (8). MC estimation using output sequences from multinomial sampling is straightforward, as one can directly sample from the original distribution $p(\boldsymbol{y} \mid \boldsymbol{x}, \boldsymbol{w})$ without the need for additional weighting or adjustment. However, the resulting estimator has high variance, and sampling only considers the likelihood and not the semantic diversity. Furthermore, multinomial sampling may miss likely output sequences that capture important

information about semantic cluster probabilities, especially if those sequences require choosing a lower-likelihood token. This limits the accurate estimation of the semantic entropy, as sampling semantically diverse output sequences essentially occurs by chance. Importance sampling, on the other hand, can overcome these limitations by incorporating semantic diversity into the sampling procedure. However, this requires a beneficial proposal distribution $q(\boldsymbol{y} \mid \boldsymbol{x}, \boldsymbol{w})$. We propose Semantically Diverse Language Generation (`SDLG`) to sample according to such an empirical proposal distribution. It seeks to efficiently explore semantic clusters, capturing important modes of $p(c \mid \boldsymbol{x}, \boldsymbol{w})$ that might be missed by multinomial sampling.

**Semantic diversity and where to find it.** Given an input sequence $\boldsymbol{x}$ and an initial output sequence $\boldsymbol{y}'$ generated by a language model, how can we generate another output sequence that has a different semantics from $\boldsymbol{y}'$? In natural language, individual words contribute to the semantics of the sentence to varying extents. For instance, consider the sentence "Einstein proposed the theory of relativity". The word "proposed" can be substituted with "introduced" without altering the semantics. However, replacing "Einstein" with "Newton" would change the semantics. Therefore, our method focuses on identifying and substituting the tokens that are most critical to the semantics of $\boldsymbol{y}'$ (see Fig. 1). This is achieved by introducing three scores at the token level that quantify each token's relevance in altering the semantics:

1. **Attribution score:** Each initial token $y_i \in \boldsymbol{y}'$ is scored by its contribution to the semantic meaning of $\boldsymbol{y}'$. We refer to this initial token's score as $A_i$.
2. **Substitution score:** Each alternative token $v_j \in \mathcal{V}$ is scored by its influence in altering the semantic meaning when substituting $y_i$ with $v_j$. We refer to this alternative token's score as $S_{ij}$.
3. **Importance score:** Each alternative token $v_j \in \mathcal{V}$ is scored by the probability the language model assigns to $v_j$ given the context up to $y_i$. We refer to this alternative token's score as $I_{ij}$.

At a high level, `SDLG` explicitly substitutes tokens based on these scores. High scores indicate a high potential to give rise to a likely output sequence with semantics different from $\boldsymbol{y}'$, as detailed below. Before that, we describe how to calculate the three scores.

Computing the attribution and substitution scores requires a loss L, which expresses to what degree $\boldsymbol{y}'$ is semantically different from itself. It is computed utilizing an NLI model that predicts whether two sequences entail or contradict each other. First, the initial output sequence is fed into the NLI model twice, resulting in a high probability of *entailment*. Second, the loss L is computed for the target prediction *contradiction*. This loss is used to compute the gradient $\nabla_{\boldsymbol{z}_i} \mathrm{L}$ w.r.t. the token embedding $\boldsymbol{z}_i$ that represents the initial token $y_i$. This gradient vector quantifies the required change in $\boldsymbol{z}_i$ to achieve a high probability of *contradiction*, thus altering the semantic meaning of $\boldsymbol{y}'$.

**Attribution score.** Our first objective is to identify which initial token $y_i$ should be changed according to the computed gradient vector. An initial token's attribution score

$$A_i \;=\; \|\boldsymbol{z}_i \odot \nabla_{\boldsymbol{z}_i} \mathrm{L}\|_2 \tag{9}$$

is defined as the Euclidean distance $\| \cdot \|_2$ of the gradient vector multiplied elementwise with the embedding vector $\boldsymbol{z}_i$ that represents the initial token $y_i$. The higher the attribution score $A_i$, the higher the impact of the token $y_i$ on altering the semantics when being changed (Adebayo et al., 2018). We note that different attribution methods could be utilized, and future work may benefit from exploring these methods to compute $A_i$ (Madsen et al., 2022).

**Substitution score.** Identifying which initial token should be changed is crucial but not sufficient on its own. We also have to identify which token to change to, as not every substitution alters the semantic meaning of the initial output sequence. Thus, our second objective is to identify appropriate alternative tokens $v_j$ that most effectively alter the semantics when substituting an initial token $y_i$. The alternative token's substitution score

$$S_{ij} \;=\; \frac{(\boldsymbol{z}_i - \boldsymbol{z}_j) \cdot \nabla_{\boldsymbol{z}_i} \mathrm{L}}{\|\boldsymbol{z}_i - \boldsymbol{z}_j\|_2 \; \|\nabla_{\boldsymbol{z}_i} \mathrm{L}\|_2} \tag{10}$$

is defined as the cosine similarity $sim(\cdot, \cdot)$ between the gradient vector and the difference between the initial token's embedding vector $\boldsymbol{z}_i$ and the alternative token's embedding vector $\boldsymbol{z}_j$. Although cosine similarity is a widely used measure for text similarity, we observed that the dot product empirically yields comparable performance. The higher the substitution score $S_{ij}$, the more closely the change in the embedding vector aligns with the direction of the gradient vector, thus the direction of altering the semantics (Mikolov et al., 2013).

**Algorithm 1** SDLG

**Output:** Semantic diverse output sequences $\mathcal{S}$
**Input:** Language model $g(\cdot)$, input sequence $\boldsymbol{x}$, vocabulary $\mathcal{V}$, number output sequences $N$
1: $\mathcal{S} \leftarrow \emptyset$
2: $\boldsymbol{y}^1 \leftarrow g(\boldsymbol{x})$
3: $\mathcal{S} \leftarrow \mathcal{S} \cup \{\boldsymbol{y}^1\}$
4: $\mathcal{R} \leftarrow$ Alg. 2
5: **for** $n = 2$ **to** $N$ **do**
6: $\quad (i, j) \leftarrow \mathcal{R}_n$
7: $\quad \boldsymbol{x}^n \leftarrow \boldsymbol{x} \oplus \boldsymbol{y}^1_{<i} \oplus v_j$
8: $\quad \boldsymbol{y}^n_{\text{rest}} \leftarrow g(\boldsymbol{x}^n)$
9: $\quad \boldsymbol{y}^n \leftarrow \boldsymbol{y}^1_{<i} \oplus v_j \oplus \boldsymbol{y}^n_{\text{rest}}$
10: $\quad \mathcal{S} \leftarrow \mathcal{S} \cup \{\boldsymbol{y}^n\}$
11: **return** $\mathcal{S}$

**Algorithm 2** Token Score Ranking

**Output:** Token pair indices ranking $\mathcal{R}$
**Input:** see Alg. 1, NLI model $e(\cdot, \cdot)$, cross-entropy loss function $l(\cdot, \cdot)$, method $Rank(\cdot)$
1: $\mathcal{R} \leftarrow \emptyset$
2: $\mathrm{L} \leftarrow l(e(\boldsymbol{y}, \boldsymbol{y}), c_{\text{contradiction}})$
3: **for** $y_i \in \boldsymbol{y}^1$ **do**
4: $\quad \nabla_{\boldsymbol{z}_i} \mathrm{L} \leftarrow \frac{\partial \mathrm{L}}{\partial y_i}$
5: $\quad A_i \leftarrow \|\boldsymbol{z}_i \odot \nabla_{\boldsymbol{z}_i} \mathrm{L}\|_2$
6: $\quad$ **for** $v_j \in \mathcal{V}$ **do**
7: $\quad\quad S_{ij} \leftarrow sim(\boldsymbol{z}_i - \boldsymbol{z}_j, \nabla_{\boldsymbol{z}_i} \mathrm{L})$
8: $\quad\quad I_{ij} \leftarrow p(v_j \mid \boldsymbol{y}^1_{<i}, \boldsymbol{x}, \boldsymbol{w})$
9: $\quad\quad \mathcal{R} \leftarrow \mathcal{R} \cup \{(A_i, S_{ij}, I_{ij})\}$
10: $\mathcal{R} \leftarrow Rank(\mathcal{R})$
11: **return** $\mathcal{R}$

Algorithm: Illustration of the workflow of SDLG, sampling diverse yet coherent output sequences by identifying and substituting semantically important tokens in the initial output sequence using three distinct context-sensitive scores.

**Importance score.** Identifying which alternative token should substitute which initial token is important but not sufficient. Not every alternative token $v_j$ is an appropriate substitution for the initial token $y_i$, given the context. The context is the input sequence $\boldsymbol{x}$ and the output sequence up to the token that is to be substituted $\boldsymbol{y}_{<i}$. Substituting with a very unlikely token might undermine the coherence of the overall sequence. Thus, our third objective is to favor alternative tokens that exhibit a high likelihood. The alternative token's importance score

$$I_{ij} = p(v_j \mid \boldsymbol{y}_{<i}, \boldsymbol{x}, \boldsymbol{w}) \tag{11}$$

is simply defined as the probability that the language model assigns to $v_j$ given the context. The higher the importance score $I_{ij}$, the more suitable the alternative token is for substitution, as it ensures that the resulting output sequence not only is semantically diverse but also remains likely.

In summary, the three scores work together to identify the appropriate initial and alternative token pair that should be substituted to alter the semantics of the initial output sequence. The *attribution* score identifies which initial token to substitute, while the *substitution* score identifies which alternative token to replace the initial token with. The *importance* score identifies which alternative token the language model considers a fit for substitution. Each of the scores positively contributes to the performance of SDLG, as detailed in Sec. E of the appendix.

**Generating semantic diverse output sequences.** It has to be noted that substituting initial tokens not corresponding to the beginning of a word is often impractical. Consequently, we exclusively apply substitutions to tokens at the beginning of a word, making them even more efficient. All token pairs $(y_i, v_j)$ are ranked according to the three scores $(A_i, S_{ij}, I_{ij})$. In its simplest form, the ranking is based on equally weighting the three scores, which we found to work well empirically. Based on this token score ranking, a new output sequence is generated by deliberately substituting the highest-ranked token pair. Subsequent tokens are discarded as they are conditioned on the substituted token, which affects their likelihood. The remainder of the new output sequence is then generated by the language model using the usual sampling strategy (see Alg. 1 and 2). SDLG preserves the semantically less relevant part of the output sequence, eliminating the need to regenerate it and focusing on the part with a high chance of altering the semantic meaning. As a result, the generation process becomes computationally more efficient than sampling from scratch each time and potentially sampling multiple duplicate output sequences.

**Proposal distribution.** Finally, we can discuss the exact form of the proposal distribution $q(\boldsymbol{y} \mid \boldsymbol{x}, \boldsymbol{w})$ in Eq. (8), which is induced by SDLG. We define the proposal distribution as

$$q(\boldsymbol{y} \mid \boldsymbol{x}, \boldsymbol{w}) = \sum_{\boldsymbol{y}' \in \mathcal{Y}} p(\boldsymbol{y} \mid \boldsymbol{y}', \boldsymbol{x}, \boldsymbol{w}) \, p(\boldsymbol{y}' \mid \boldsymbol{x}, \boldsymbol{w}) . \tag{12}$$

The probability distribution $p(\boldsymbol{y} \mid \boldsymbol{y}', \boldsymbol{x}, \boldsymbol{w})$ denotes SDLG transforming a given output sequence $\boldsymbol{y}'$ into another output sequence $\boldsymbol{y}$ with a high chance of changing the semantics as well. However,

since $\boldsymbol{y}'$ is not known a priori, we sum over all possible $\boldsymbol{y}'$ according to their probability of being sampled $p(\boldsymbol{y}' \mid \boldsymbol{x}, \boldsymbol{w})$. Under assumptions about which index $t$ is likely to be chosen for a given $\boldsymbol{y}'$, the proposal distribution takes the form

$$q(\boldsymbol{y} \mid \boldsymbol{x}, \boldsymbol{w}) \;=\; \frac{p(\boldsymbol{y} \mid \boldsymbol{x}, \boldsymbol{w})}{p(y_t \mid \boldsymbol{y}_{<t}, \boldsymbol{x}, \boldsymbol{w})} \tag{13}$$

where $p(y_t \mid \boldsymbol{y}_{<t}, \boldsymbol{x}, \boldsymbol{w})$ is the likelihood of the token that is exchanged by SDLG. Intuitively, it means that we have to adjust the MC estimate in Eq. (6), because SDLG interferes in sampling and changes the token $y_t$ deterministically. For more details on the assumptions and a step-by-step derivation, see Sec. B in the appendix. Our experiments show that sampling according to this proposal distribution improves uncertainty estimation in NLG.

## 5 RELATED WORK

**Uncertainty estimation in NLG.** Several works utilized the language model itself to obtain a prediction of their uncertainty, whether that be numerical or verbal (Mielke et al., 2022; Lin et al., 2022b; Kadavath et al., 2022; Cohen et al., 2023a; Ganguli et al., 2023; Ren et al., 2023; Tian et al., 2023). Cohen et al. (2023b) utilize cross-examination, where one language model generates the output sequence, and the other language model acts as an examiner to assess the uncertainty. Zhou et al. (2023) investigate the behavior of language models when expressing their (un)certainty.

A large body of work focuses on sampling a set of output sequences to obtain sampling-based uncertainty estimators. Xiao and Wang (2021); Malinin and Gales (2021); Hou et al. (2023) incorporate both aleatoric and epistemic estimates of uncertainty, where epistemic uncertainty due to model selection is considered. While Kuhn et al. (2023); Duan et al. (2023); Farquhar et al. (2024) evaluate the aleatoric uncertainty only under a single given language model, they take the semantic equivalence of potential output sequences into account. Manakul et al. (2023) also sample a set of output sequences but utilize them as input to another language model to assess the uncertainty. Another approach to uncertainty estimation in NLG is conformal prediction (Quach et al., 2023), where a stopping rule for generating output sequences is calibrated. Additionally, Xiao et al. (2022) empirically analyze how factors such as model architecture and training details influence the uncertainty estimates in language models.

**Complementary methods.** Related work on uncertainty estimation in NLG aims to propose improved uncertainty measures compared to semantic entropy and could benefit from more diverse samples. For instance, Lin et al. (2023) propose leveraging the similarity between output sequences, while Chen et al. (2024) introduce EigenScore, which utilizes embeddings of output sequences. These approaches could integrate SDLG to generate the output sequences needed to compute their uncertainty estimators, making them complementary to SDLG.

**Generating diverse output sequences.** Li et al. (2016) propose an alternative training procedure of language models to avoid generic, input-independent output sequences and increase diversity. Diverse beam search (Vijayakumar et al., 2018) optimizes for a diversity-augmented objective across beam groups, based on diversity heuristics. Ippolito et al. (2019) compare diversity-encouraging decoding strategies. Nucleus sampling (Holtzman et al., 2020) generates higher quality as well as more diverse output sequences but does not explicitly encourage semantic diversity. Contrastive decoding (Li et al., 2023) utilizes a second, weaker language model, where the decoding algorithm favors tokens generated by the stronger model and penalizes tokens generated by the weaker model. Tam (2020) utilize semantic clustering during beam search, which is used to prune beams and diversify the remaining candidates. However, this only indirectly steers towards more diversity and relies on the diversity of the initial beams.

Closely related, but not directly targeting the semantic diversity of output sequences, is the field of (neural) controllable text generation (Prabhumoye et al., 2020). Here, the generation process of the language model is steered by another language model to e.g. adhere to a certain dialog structure, prevent toxic answers, or play a certain persona. Keskar et al. (2019) use control codes added to the prompt to steer the generation. Dathathri et al. (2020) propose the use of an external supervised classifier to control the generation. Chan et al. (2021) also utilize an external classifier but train in a self-supervised setting. (Ghazvininejad et al., 2017; Holtzman et al., 2018) re-weight the probability distributions at each step of generating the output sequence. For further work in this field, see the surveys by Prabhumoye et al. (2020); Zhang et al. (2023).

Table 1: AUROC using different uncertainty measures as a score to distinguish between correct and incorrect answers. $SE_{MS}$ uses the improper semantic entropy estimator implemented by Kuhn et al. (2023) while $SE^{(*)}_{...}$ use the proper semantic entropy estimator as we introduced in Sec. 3, with different sampling strategies. The threshold of the correctness metric Rouge-L (F1 score) is set to 0.5. Each method uses ten output sequences for assigning an uncertainty estimate.

| Dataset | # Param. | LN-PE | PE | SAR | $SE_{MS}$ | $SE^{(*)}_{MS}$ | $SE^{(*)}_{DBS}$ | $SE^{(*)}_{SDLG}$ |
|---|---|---|---|---|---|---|---|---|
| **TruthfulQA** | **2.7b** | .439 | .517 | .611 | .405 | .846 | .686 | **.920** |
| | **6.7b** | .446 | .510 | .555 | .512 | .781 | .637 | **.881** |
| | **13b** | .676 | .712 | .775 | .453 | .896 | .819 | **.956** |
| | **30b** | .482 | .542 | .517 | .438 | .864 | .788 | **.927** |
| **CoQA** | **2.7b** | .717 | .693 | .733 | .717 | **.744** | .697 | **.744** |
| | **6.7b** | .728 | .703 | .748 | .739 | **.764** | .714 | .759 |
| | **13b** | .723 | .697 | .747 | .743 | .758 | .720 | **.760** |
| | **30b** | .732 | .698 | .742 | .745 | .767 | .713 | **.768** |
| **TriviaQA** | **2.7b** | .769 | .787 | .785 | .781 | .804 | .808 | **.809** |
| | **6.7b** | .790 | .805 | .804 | .803 | .822 | .823 | **.829** |
| | **13b** | .807 | .820 | .819 | .824 | .838 | .841 | **.845** |
| | **30b** | .799 | .812 | .815 | .817 | .831 | .837 | **.840** |

## 6 EXPERIMENTS

**Data and models.** To ensure a fair comparison of different uncertainty estimation methods within the scope of our computational budget, we decided to align the evaluation tasks with current work by focusing on free-form question answering. Thus, we performed experiments on three datasets that cover a broad range of question answering settings. To be concrete, we use the over 800 closed-book questions in TruthfulQA (Lin et al., 2022a) corresponding to whole sentence answers, the almost 8,000 open-book questions in the development split of CoQA (Reddy et al., 2019) corresponding to medium to shorter length answers, and about 8,000 closed-book questions in the training split of TriviaQA (Joshi et al., 2017) corresponding to short, precise answers. We use a 5-shot, zero-shot, and 10-shot prompt for TruthfulQA, CoQA, and TriviaQA, respectively. These datasets are frequently used as benchmarks for uncertainty estimation in NLG due to their strong correlation with human evaluations and the effective performance of "off-the-shelf" language models compared to tasks requiring fine-tuning (Kuhn et al., 2023; Goyal et al., 2023). Each of the three datasets was evaluated with the OPT model family (Zhang et al., 2022), with model sizes ranging from 2.7 to 30 billion parameters. Related work suggests that performance trends generalize across transformer-based model families (Duan et al., 2023; Manakul et al., 2023). In general, the four language models and three datasets assess the performance of uncertainty estimation methods in NLG across varying model sizes, output sequence lengths, and both open-book and closed-book settings.

**Evaluation.** Effective uncertainty measures should accurately reflect the reliability of answers generated by the language model. Higher uncertainty more likely leads to incorrect generations. Thus, to evaluate the performance of an uncertainty estimator, we assess how well it correlates with the correctness of the language model's answers; correct answers should be assigned a lower uncertainty estimator than incorrect answers. To determine whether an answer is correct, it has to be compared to the respective ground truth answer. First, we utilize the statistics-based metrics Rouge-L and Rouge-1 (Lin, 2004), which take the initial output sequence and the ground truth answer to measure the longest common subsequence and the overlap of unigrams, respectively. Second, we utilize the learning-based metric BLEURT (Sellam et al., 2020) that uses a learned evaluation model to assess how well the initial output sequence conveys the meaning of the ground truth answer. It has to be noted that for each of these three metrics, correctness is computed as the difference between the maximum score assigned to a true reference answer and the maximum score assigned to a false reference answer. While TruthfulQA provides both true and false reference answers, CoQA and TriviaQA only provide true reference answers. The resulting scalar value reflects the correctness of the answer and is subsequently thresholded to yield a binary decision. To ensure a robust evaluation, we use a comprehensive set of ten correctness thresholds ranging from 0.1 to 1.0 (exact match). To measure the correlation between incorrectness of answers and the respective uncertainty estimates, we utilize AUROC, with higher values indicating better performance of the uncertainty estimator, as

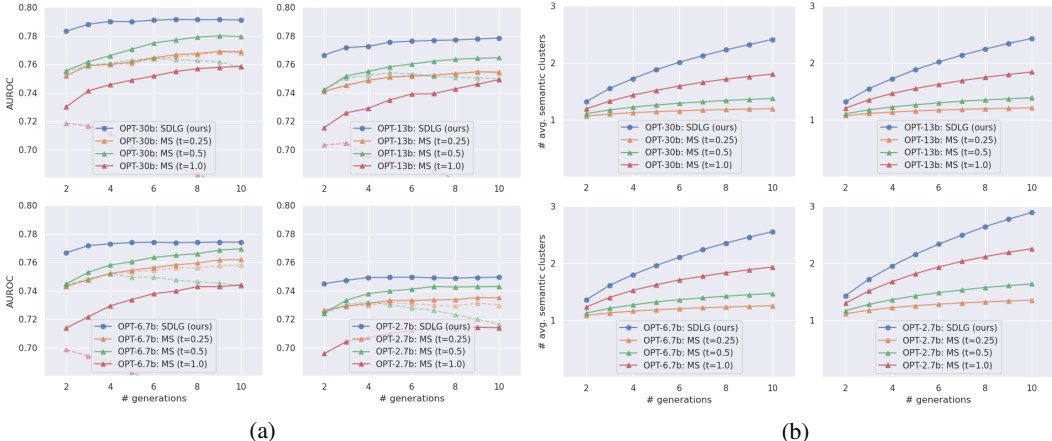

Figure 3: (a) AUROC using uncertainty measures across various numbers of samples as score to distinguish between correct and incorrect answers of the CoQA dataset. Solid and dotted lines indicate the performance when using the proper and improper semantic entropy estimator, respectively. (b) Average number of semantic clusters found across various numbers of samples considered.

it reflects a stronger alignment between the correctness of the language model's answers and their respective uncertainty estimates. Overall, this evaluation process follows established methodologies for assessing the performance of uncertainty measures in NLG (Kuhn et al., 2023; Lin et al., 2023; Duan et al., 2023; Bakman et al., 2024; Farquhar et al., 2024; Nikitin et al., 2024).

**Baselines.** We compare our method against methods that directly utilize the predictive entropy of the output distribution on a token level. These are Predictive Entropy (PE), Length-Normalized Predictive Entropy (LN-PE) (Malinin and Gales, 2021), and Shifting Attention to Relevance (SAR) (Duan et al., 2023). We also compare against methods that utilize semantic entropy on a sequence level. Thereby, output sequences are generated with multinomial sampling ($SE_{MS}$) (Kuhn et al., 2023) or with diverse beam search ($SE_{DBS}$) (Vijayakumar et al., 2018). Although DBS has not explicitly been proposed for uncertainty estimation in NLG, we added it as a more traditional sampling method that enforces diversity among output sequences.

**Our method (`SDLG`).** Unlike current methods that rely on finding the optimal sampling temperature or penalty term, SDLG does not require hyperparameter tuning as it controls the sampling. One has only to decide on the ranking method for the three individual token scores. We empirically found that the performance of our method is quite robust with respect to the weighting of the token scores. Therefore, throughout the experiments, we derive the final token score ranking by straightforwardly averaging the three individual token scores. To compute the token scores for semantic diversity as discussed in Sec. 4, we utilize the same NLI model DeBERTa (Williams et al., 2018; He et al., 2021) that is also used for predicting semantic equivalences and determining semantic clusters.

**Analysis of results.** Tab. 1 summarizes the main results of our experiments. It can be observed that SDLG largely outperforms all baselines throughout datasets and model sizes. Tab. 2 together with Fig. 6 - 8 in the appendix show that the performance improvements of our method persist across the three correctness metrics Rouge-L (F1 score), Rouge-1 (F1 score), and BLEURT, as well as different correctness thresholds and the number of samples considered. The results show that using the proper semantic entropy estimator for the semantic entropy ($SE_{MS}^{(*)}$) outperforms the improper estimator ($SE_{MS}$) when generating output sequences via MS, as outlined in Sec. 3.

It can be observed that simple token-level diversity enforced by higher temperatures in MS (Kuhn et al., 2023) or by DBS (Vijayakumar et al., 2018) is insufficient for capturing semantic diversity essential for uncertainty estimation in NLG. However, since TriviaQA includes the shortest output sequences among the datasets, the advantages of our method are less pronounced. Although our method still outperforms the baseline methods on short output sequences, the true strengths of SDLG emerge in scenarios with longer output sequences, such as those in TruthfulQA, where simple token-level diversity is insufficient, and the ability of our method to explore semantic diversity on a sequence-level becomes critical. The smallest performance gap between SE with MS and

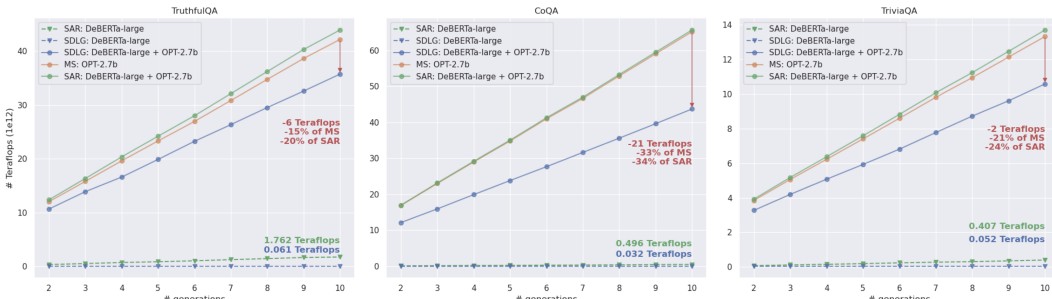

Figure 4: Average number of Teraflops required for an increasing number of samples generated with `SDLG` vs. standard multinomial sampling (MS) and Shifting Attention to Relevance (SAR).

`SDLG` occurs on the CoQA dataset. Therefore, we more closely examine the relationship between the performance and the sample size. We report MS with the optimal temperature ($t = 0.5$) as well as at nearby temperatures for comparison. As shown in Fig. 3a, while the performance difference is smaller when considering all ten generations, `SDLG` demonstrates strong performance even after sampling just one additional output sequence. This highlights the sample efficiency of our method.

**Semantic clusters.** Fig. 3b shows that our method results in a $19\%$ increase of semantic clusters after generating the second output sequence, as well as a $74\%$ increase of semantic clusters after the tenth output sequence, compared to multinomial sampling with the highest-performing temperature used by the current state-of-the-art method (Kuhn et al., 2023), when averaging across all CoQA instances. This is because `SDLG` explicitly searches for output sequences with different semantic meanings and practically does not sample the same output sequence twice.

**Computational expenses.** While it is true that generating additional output sequences with `SDLG` requires extra computation for the NLI model to obtain the token score ranking, the primary computational effort lies in the generation of output sequences. The computational expenses of a single forward and backward pass through the NLI model with a few hundred million parameters is minor compared to generating only a single token using a language model with billions of parameters. Since our method deterministically changes a specific token within the initial output sequence, preceding tokens do not have to be regenerated again, but only subsequent ones. This results in `SDLG` requiring at least an average of $15\%$ (TruthfulQA), $33\%$ (CoQA), and $21\%$ (TriviaQA) fewer flops compared to multinomial sampling that is used by the current state-of-the-art method (Kuhn et al., 2023). In general, the advantage of our method over the current methods further increases with longer output sequences and larger language model sizes, as illustrated in Fig. 4.

## 7 CONCLUSION

We introduce `SDLG` to improve uncertainty estimation in NLG. `SDLG` substitutes tokens in the initial output sequence that are likely to lead to a change in semantic meaning. Unlike standard multinomial sampling, this targeted approach effectively samples likely output sequences with different semantic meanings, capturing important information for estimating semantic uncertainty. Our experiments on free-form question answering datasets demonstrate that `SDLG` not only increases the overall quality of the uncertainty estimator but is also computationally more efficient.

Future work should address the assumption that each sentence can be attributed to a single semantic cluster, which may be overly restrictive. This represents an orthogonal issue that does not conflict with our method but can be addressed independently. Also, this work focuses on estimating the aleatoric semantic uncertainty. Future work should investigate how to effectively assess epistemic semantic uncertainty, which is a challenging task in itself (see Eq. (3)).

Despite these remaining challenges, `SDLG` already offers notable advantages over prior methods. First, `SDLG` controls the sampling instead of depending on chance to obtain diverse samples. It eliminates the necessity of searching for an optimal sampling temperature, an important hyperparameter for all prior methods. Second, the advantageous sampling reduces the required number of samples for high-quality uncertainty estimation, while also being computationally most efficient per sample. Overall, `SDLG` considerably enhances the applicability of uncertainty estimation in NLG.

## ETHICS AND REPRODUCIBILITY STATEMENT

We recognize the potential for language models to generate biased or harmful content if not properly managed. While `SDLG` aims to improve model reliability, we strongly advocate for its responsible use alongside bias mitigation strategies and content moderation techniques to ensure ethical and safe deployment.

To promote reproducibility, theoretical justifications are provided in Sec. 2, Sec. 3, and Sec. 4, with further mathematical derivations and proofs in the appendix. All datasets used are publicly available, and we utilize standard benchmarks to facilitate easy replication of our work. Upon publication, the source code for reproducing all experiments will be made publicly accessible.

## ACKNOWLEDGEMENTS

The ELLIS Unit Linz, the LIT AI Lab, the Institute for Machine Learning, are supported by the Federal State Upper Austria. This research was funded in part by the Austrian Science Fund (FWF) [10.55776/COE12]. We thank the projects INCONTROL-RL (FFG-881064), PRIMAL (FFG-873979), S3AI (FFG-872172), DL for GranularFlow (FFG-871302), EPILEPSIA (FFG-892171), FWF AIRI FG 9-N (10.55776/FG9), AI4GreenHeatingGrids (FFG- 899943), INTE-GRATE (FFG-892418), ELISE (H2020-ICT-2019-3 ID: 951847), Stars4Waters (HORIZON-CL6-2021-CLIMATE-01-01). We thank NXAI GmbH, Audi.JKU Deep Learning Center, TGW LO-GISTICS GROUP GMBH, Silicon Austria Labs (SAL), FILL Gesellschaft mbH, Anyline GmbH, Google, ZF Friedrichshafen AG, Robert Bosch GmbH, UCB Biopharma SRL, Merck Healthcare KGaA, Verbund AG, GLS (Univ. Waterloo), Software Competence Center Hagenberg GmbH, Borealis AG, TÜV Austria, Frauscher Sensonic, TRUMPF and the NVIDIA Corporation. Kajetan Schweighofer acknowledges travel support from ELISE (GA no 951847).

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

## A   BROADER IMPACT

This work focuses on assessing the uncertainty in natural language generation (NLG) using language models. Our primary goal is to increase the robustness of language models, assess the reliability of their predicted output sequences, and detect when a language model is hallucinating. Therefore, we contend that our work makes a positive contribution to society in several aspects:

1. Improved discernment of certainty in model predictions enhances practical application in real-world scenarios. This can be implemented by signaling uncertainty to users, such as through highlighting dubious sections of responses or opting not to display uncertain outputs altogether.
2. Reliable uncertainty estimates may increase the trust of the user in the language model, as it provides a basis to gauge the quality of the answer.

However, while we expect mainly a positive impact on society, there are also potential negative aspects:

1. Enhanced uncertainty estimation might not yield expected outcomes if users lack the necessary training to interpret these estimates effectively.
2. While better uncertainty assessment can foster usability and user trust, it also carries the risk of creating undue reliance on these models. It is crucial to maintain human oversight and critical evaluation of language model outputs, as over-reliance can be detrimental.

It is important to note that our method evaluates uncertainty based on the information available to the language model. Therefore, it may inaccurately deem a factually incorrect answer as certain if the model is consistently trained on such erroneous facts. This issue, often perceived as model "hallucination", is not a reflection of the model's uncertainty, but rather a result of factual inaccuracies in the underlying data that is additional to the hallucinations stemming from uncertainty.

## B   ON THE PROPOSAL DISTRIBUTION INDUCED BY SDLG

In the following, we analyze the proposal distribution induced by SDLG. We consider a probabilistic transformation of one output sequence $\boldsymbol{y}'$ into another output sequence $\boldsymbol{y}$, given by $p(\boldsymbol{y} \mid \boldsymbol{y}', \boldsymbol{x}, \boldsymbol{w})$. This is introduced because we have to sum over all possible output sequences $\boldsymbol{y}'$ that we could apply SDLG on, leading to

$$q(\boldsymbol{y} \mid \boldsymbol{x}, \boldsymbol{w}) \;=\; \sum_{\boldsymbol{y}' \in \mathcal{Y}} p(\boldsymbol{y}' \mid \boldsymbol{x}, \boldsymbol{w})\, p(\boldsymbol{y} \mid \boldsymbol{y}', \boldsymbol{x}, \boldsymbol{w}) \,. \tag{14}$$

We can write $p(\boldsymbol{y} \mid \boldsymbol{y}', \boldsymbol{x}, \boldsymbol{w})$ as an expected value over $t$, the index where SDLG chooses a different token:

$$p(\boldsymbol{y} \mid \boldsymbol{y}', \boldsymbol{x}, \boldsymbol{w}) \;=\; \sum_{t=1}^{T} p(t \mid \boldsymbol{y}', \boldsymbol{x}, \boldsymbol{w})\, p(\boldsymbol{y} \mid t, \boldsymbol{y}', \boldsymbol{x}, \boldsymbol{w}) \,. \tag{15}$$

The construction of $\boldsymbol{y}$ from $\boldsymbol{y}'$ only changes one element of $\boldsymbol{y}'$ at position $t$ and then generates the new postfix. Therefore, we have $p(\boldsymbol{y} \mid t, \boldsymbol{y}', \boldsymbol{x}, \boldsymbol{w}) = 0$ for $\boldsymbol{y}'_{<t} \neq \boldsymbol{y}_{<t}$. Consequently, $\sum_{\boldsymbol{y}' \in \mathcal{Y}} p(\boldsymbol{y}' \mid \boldsymbol{x}, \boldsymbol{w})$ can be reduced to $\sum_{\boldsymbol{y}'_{>t} \in \mathcal{Y}_{>t}} p(\boldsymbol{y}'_{>t} \mid \boldsymbol{y}_{\leqslant t}, \boldsymbol{x}, \boldsymbol{w})\, p(\boldsymbol{y}_{\leqslant t} \mid \boldsymbol{x}, \boldsymbol{w})$ if the factor $p(\boldsymbol{y} \mid t, \boldsymbol{y}', \boldsymbol{x}, \boldsymbol{w})$ is present. There is only one possibility for the prefix with $\boldsymbol{y}'_{<t} = \boldsymbol{y}_{<t}$.

Using Eq. (15) in Eq. (14) leads to

$$q(\boldsymbol{y} \mid \boldsymbol{x}, \boldsymbol{w}) \tag{16}$$

$$= \sum_{\boldsymbol{y}' \in \mathcal{Y}} p(\boldsymbol{y}' \mid \boldsymbol{x}, \boldsymbol{w}) \sum_{t=1}^{T} p(t \mid \boldsymbol{y}', \boldsymbol{x}, \boldsymbol{w})\, p(\boldsymbol{y} \mid t, \boldsymbol{y}', \boldsymbol{x}, \boldsymbol{w})$$

$$= \sum_{t=1}^{T} \sum_{\boldsymbol{y}' \in \mathcal{Y}} p(\boldsymbol{y}' \mid \boldsymbol{x}, \boldsymbol{w})\, p(t \mid \boldsymbol{y}', \boldsymbol{x}, \boldsymbol{w})\, p(\boldsymbol{y} \mid t, \boldsymbol{y}', \boldsymbol{x}, \boldsymbol{w})$$

$$= \sum_{t=1}^{T} \sum_{\boldsymbol{y}'_{>t} \in \mathcal{Y}_{>t}} \sum_{\boldsymbol{y}'_{\leqslant t} \in \mathcal{Y}_{\leqslant t}} p(\boldsymbol{y}'_{>t} \mid \boldsymbol{y}'_{\leqslant t}, \boldsymbol{x}, \boldsymbol{w})\, p(\boldsymbol{y}'_{\leqslant t} \mid \boldsymbol{x}, \boldsymbol{w})\, p(t \mid \boldsymbol{y}', \boldsymbol{x}, \boldsymbol{w})\, p(\boldsymbol{y} \mid t, \boldsymbol{y}', \boldsymbol{x}, \boldsymbol{w})$$

$$= \sum_{t=1}^{T} \sum_{\boldsymbol{y}'_{>t} \in \mathcal{Y}_{>t}} p(\boldsymbol{y}'_{>t} \mid \boldsymbol{y}_{\leqslant t}, \boldsymbol{x}, \boldsymbol{w})\, p(\boldsymbol{y}_{\leqslant t} \mid \boldsymbol{x}, \boldsymbol{w})\, p(t \mid \boldsymbol{y}'_{>t}, \boldsymbol{y}_{\leqslant t}, \boldsymbol{x}, \boldsymbol{w})\, p(\boldsymbol{y}_{>t} \mid \boldsymbol{y}'_{>t}, \boldsymbol{y}_{\leqslant t}, \boldsymbol{x}, \boldsymbol{w})$$

$$= \sum_{t=1}^{T} \sum_{\boldsymbol{y}'_{>t} \in \mathcal{Y}_{>t}} p(\boldsymbol{y}'_{>t} \mid \boldsymbol{y}_{\leqslant t}, \boldsymbol{x}, \boldsymbol{w})\, p(\boldsymbol{y}_{\leqslant t} \mid \boldsymbol{x}, \boldsymbol{w})\, p(t \mid \boldsymbol{y}'_{>t}, \boldsymbol{y}_{\leqslant t}, \boldsymbol{x}, \boldsymbol{w})\, p(\boldsymbol{y}_{>t} \mid \boldsymbol{y}_{\leqslant t}, \boldsymbol{x}, \boldsymbol{w})$$

$$= \sum_{t=1}^{T} p(\boldsymbol{y}_{\leqslant t} \mid \boldsymbol{x}, \boldsymbol{w}) \left( \sum_{\boldsymbol{y}'_{>t} \in \mathcal{Y}_{>t}} p(\boldsymbol{y}'_{>t} \mid \boldsymbol{y}_{\leqslant t}, \boldsymbol{x}, \boldsymbol{w})\, p(t \mid \boldsymbol{y}'_{>t}, \boldsymbol{y}_{\leqslant t}, \boldsymbol{x}, \boldsymbol{w}) \right) p(\boldsymbol{y}_{>t} \mid \boldsymbol{y}_{\leqslant t}, \boldsymbol{x}, \boldsymbol{w})$$

$$= \sum_{t=1}^{T} p(\boldsymbol{y}_{\leqslant t} \mid \boldsymbol{x}, \boldsymbol{w})\, p(t \mid \boldsymbol{y}_{\leqslant t}, \boldsymbol{x}, \boldsymbol{w})\, p(\boldsymbol{y}_{>t} \mid \boldsymbol{y}_{\leqslant t}, \boldsymbol{x}, \boldsymbol{w})$$

$$= \sum_{t=1}^{T} p(\boldsymbol{y}_{<t} \mid \boldsymbol{x}, \boldsymbol{w})\, p(y_t \mid \boldsymbol{y}_{<t}, \boldsymbol{x}, \boldsymbol{w})\, p(t \mid y_t, \boldsymbol{y}_{<t}, \boldsymbol{x}, \boldsymbol{w})\, p(\boldsymbol{y}_{>t} \mid y_t, \boldsymbol{y}_{<t}, \boldsymbol{x}, \boldsymbol{w})$$

$$= \sum_{t=1}^{T} p(t \mid y_t, \boldsymbol{y}_{<t}, \boldsymbol{x}, \boldsymbol{w})\, p(\boldsymbol{y}_{<t} \mid \boldsymbol{x}, \boldsymbol{w})\, p(\boldsymbol{y}_{>t} \mid y_t, \boldsymbol{y}_{<t}, \boldsymbol{x}, \boldsymbol{w}) \,,$$

where we used $p(y_t \mid \boldsymbol{y}_{<t}, \boldsymbol{x}, \boldsymbol{w}) = 1$, since SDLG chooses $y_t$ deterministically given $\boldsymbol{y}_{<t} = \boldsymbol{y}'_{<t}$. We assume that all probability mass in $p(t \mid y_t, \boldsymbol{y}_{<t}, \boldsymbol{x}, \boldsymbol{w})$ is at the actually observed $t$. This means, given all possible $\boldsymbol{y}'_{>t}$, $t$ is the most probable position to induce a semantic change. This is a strong assumption that needs further investigation in future work. Under this assumption, the final result in Eq. (16) reduces to

$$q(\boldsymbol{y} \mid \boldsymbol{x}, \boldsymbol{w}) \;=\; p(\boldsymbol{y}_{<t} \mid \boldsymbol{x}, \boldsymbol{w})\, p(\boldsymbol{y}_{>t} \mid y_t, \boldsymbol{y}_{<t}, \boldsymbol{x}, \boldsymbol{w}) \,. \tag{17}$$

We can re-write Eq. (17) in terms of the output sequence probability distribution $p(\boldsymbol{y} \mid \boldsymbol{x}, \boldsymbol{w})$ as

$$q(\boldsymbol{y} \mid \boldsymbol{x}, \boldsymbol{w}) \;=\; \frac{p(\boldsymbol{y} \mid \boldsymbol{x}, \boldsymbol{w})}{p(y_t \mid \boldsymbol{y}_{<t}, \boldsymbol{x}, \boldsymbol{w})} \,. \tag{18}$$

## C  ESTIMATING THE SEMANTIC ENTROPY

In the following, we provide further details about estimating the aleatoric semantic uncertainty, namely the semantic entropy.

### C.1  SEMANTIC ENTROPY ESTIMATOR (KUHN ET AL. 2023)

As already established in the main paper, directly using the estimator for semantic entropy in Eq. (5) is not possible because the distribution $p(c \mid \boldsymbol{x}, \boldsymbol{w})$ is not known. Inspecting the implementation of Kuhn et al. (2023) reveals that their estimator of the semantic entropy is using the estimate of the semantic cluster probability distribution

$$p(c \mid \boldsymbol{x}, \boldsymbol{w}) \approx \sum_{n=1}^{N} \mathbb{1}\{\boldsymbol{y} \in c \mid \boldsymbol{x}, \boldsymbol{w}\}\, p(\boldsymbol{y}^n \mid \boldsymbol{x}, \boldsymbol{w}) \,, \quad \boldsymbol{y}^n \sim p(\boldsymbol{y} \mid \boldsymbol{x}, \boldsymbol{w}) \,. \tag{19}$$

They then approximate the semantic entropy as

$$\mathrm{H}(p(c \mid \boldsymbol{x}, \boldsymbol{w})) \;\approx\; -\frac{1}{M} \sum_{m=1}^{M} \log p(c_m \mid \boldsymbol{x}, \boldsymbol{w}) \,. \tag{20}$$

This assumes that $c_m$ are sampled according to $p(c \mid \boldsymbol{x}, \boldsymbol{w})$, but they are not!

Eq. (20) would be a correct estimator of the semantic entropy if the semantic clusters were sampled according to $p(c_m \mid \boldsymbol{x}, \boldsymbol{w})$. However, this distribution cannot be sampled directly and Kuhn et al. (2023) utilize it to compute the semantic entropy from the class estimates Eq. (19) instead of using it as an estimator. In this scenario, Eq.(7) must be used instead. Note that it is necessary to normalize $p(c \mid \boldsymbol{x}, \boldsymbol{w})$ because the estimate is generally not a normalized probability distribution.

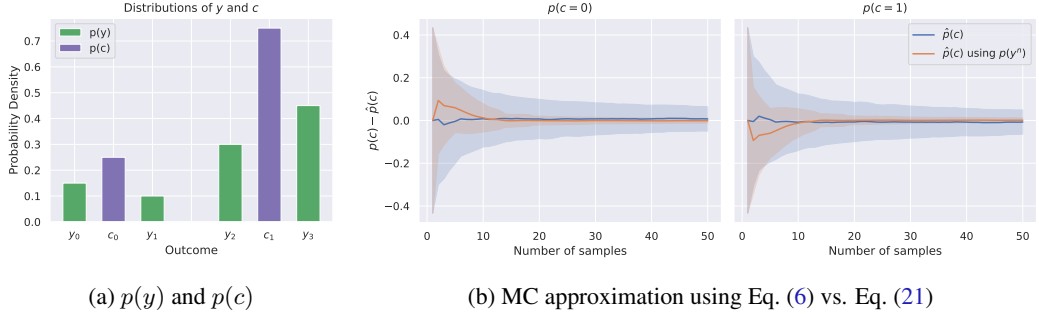

(a) $p(y)$ and $p(c)$          (b) MC approximation using Eq. (6) vs. Eq. (21)

Figure 5: Synthetic example of approximating a cluster distribution $p(c)$ of an underlying probability distribution $p(y)$. In (a) the distributions are shown. In (b), the bias and variance over 200 runs of the MC approximations per number of samples using Eq. (6) (blue) and Eq. (21) (orange) are given.

## C.2 DETAILS ON OUR SEMANTIC ENTROPY ESTIMATOR

The estimator of the semantic cluster probability distribution of Kuhn et al. (2023) given by Eq. (19) can be interpreted as Eq. (6) with importance sampling. Formally, they utilize an empirical proposal distribution $\hat{q}(\boldsymbol{y} \mid \boldsymbol{x}, \boldsymbol{w}) := \frac{1}{N} \sum_{n=1}^{N} \mathbb{1}\{\boldsymbol{y} = \boldsymbol{y}^n\}$, defined by the set of previously sampled output sequences $\{\boldsymbol{y}^n\}_{n=1}^{N}$. The approximation of the semantic cluster probability distribution given by Eq. (6) thus changes to

$$ p(c \mid \boldsymbol{x}, \boldsymbol{w}) \approx \frac{1}{N} \sum_{n=1}^{N} \mathbb{1}\{\boldsymbol{y} \in c \mid \boldsymbol{x}, \boldsymbol{w}\} \frac{p(\boldsymbol{y}^n \mid \boldsymbol{x}, \boldsymbol{w})}{\hat{q}(\boldsymbol{y}^n \mid \boldsymbol{x}, \boldsymbol{w})}, \quad \boldsymbol{y}^n \sim \hat{q}(\boldsymbol{y} \mid \boldsymbol{x}, \boldsymbol{w}). \qquad (21) $$

As this distribution is known by design and can be enumerated, Eq. (21) simplifies to a weighted sum. The quality of this approximator strongly depends on the empirical distribution. Therefore, Eq. (21) should only be used in favor of Eq. (6) if $\{\boldsymbol{y}^n\}_{n=1}^{N}$ containts output sequences that have very high probability under $p(\boldsymbol{y} \mid \boldsymbol{x}, \boldsymbol{w})$. The more these distributions differ, the higher the variance of the estimator, and therefore, the lower the approximation quality. We utilized Eq. (21) instead of Eq. (6) for the baseline using multinomial sampling and in addition to the importance sampling we perform with SDLG (c.f. Eq.(8)).

To illustrate the validity of using the estimator in Eq. (21), consider the following example: Given are a probability distribution $p(y) = (0.15, 0.1, 0.3, 0.45)$. Furthermore, the cluster probability distribution $p(c) = (0.25, 0.75)$ is derived from this distribution, thus $y_0, y_1 \in c_0$ and $y_2, y_3 \in c_1$. The distributions are shown in Fig. 5a. In Fig. 5b, we compare the MC approximation using Eq. (6) and Eq. (21). The results show that the estimator using Eq. (21) is prone to be more biased for a low number of samples but decreases its variance much faster.

Furthermore, we found that the logarithm of the unnormalized probability estimator together with normalizing the probability estimator outside the logarithm in Eq. (7) improves empirical results for all methods that estimate the semantic entropy.

## D    FURTHER EXPERIMENTS AND DETAILS

The code and data are available at https://github.com/ml-jku/SDLG.

**Hyperparameters.** For baseline methods, we performed an extensive hyperparameter search for each dataset with the $\text{SE}_{MS}$ temperature $\in \{0.25, 0.5, 1.0, 1.5, 2.0\}$ and the $\text{SE}_{DBS}$ penalty term $\in \{0.2, 0.5, 1.0\}$. Also, each method uses 10 generations to assign an uncertainty estimator, as prior work suggests that sample sizes above 10 do not significantly improve the performance of the uncertainty measures (Kuhn et al., 2023; Duan et al., 2023).

**Results.** A comprehensive overview of all conducted experiments is given in Tab. 2. The results of our reimplementation of methods PE, LN-PE, and SAR match closely with the results reported in prior work (Kuhn et al., 2023; Duan et al., 2023). Furthermore, we did an in-depth comparison of the two best semantic entropy estimators $\text{SE}_{MS}^{(*)}$ and $\text{SE}_{\text{SDLG}}^{(*)}$ for all three considered correctness metrics using an extensive range of correctness thresholds, as well as over the number of sampled output sequences. Results on the three considered datasets for all models are depicted in Fig. 6 - 8.

**Computational expenses.** The computational expenses of SDLG and previous methods are shown in Fig. 4 across a number of samples generated by a 2.7 billion parameter language model. It can be observed that the advantage of our method in terms of computational efficiency further increases with an increasing number of samples and longer input and output sequences. To further reduce the computational expenses of our method, we decrease the number of token scores that have to be computed by implementing a token probability threshold of $0.001$, under the rationale that tokens falling below this probability threshold would, in any case, be assigned a very low importance score.

**Implementational details of SDLG.** In general, the NLI model might not build upon the same token embedding or even the same vocabulary $\mathcal{V}$ as the language model. Consequently, the embedding vectors need to be differentiably transformed to enable the computation of the gradients with respect to the token embeddings. Fortunately, there exist efficient exact methods to learn the optimal linear transformation between the two monolingual embedding spaces (Artetxe et al., 2016). However, for our considered NLI and LLM models (He et al., 2021; Zhang et al., 2022), tokenizers had the same vocabulary.

## E    INSIGHTS INTO SDLG

**Illustrative example.** Fig. 10 considers the input sequence "Who proposed the theory of relativity?" with the given output sequence "Albert Einstein did". When investigating the alternative tokens it becomes clear that not every substitution leads to a change in semantic meaning. It is important to substitute tokens that also receive a high score for altering the semantics. In this example, it is the initial token corresponding to "Einstein" and the alternative token corresponding to "Schweitzer". Yet, this alone does not directly indicate a high level of uncertainty about the output sequence. High uncertainty should be attributed only if the new output sequence is completed and still has a different semantic meaning. If the language model is uncertain about the originator of the theory of relativity, it completes the new output sequence like "Albert Schweitzer proposed the theory of relativity". This would suggest a high uncertainty estimate. However, if the language model is confident about the originator of the theory of relativity, it completes the new output sequence like "Albert Schweitzer didn't, but Albert Einstein did". It is in favor of a low uncertainty estimate since the model reinforces the original semantics. This illustrates that solely considering the predictive uncertainty on a token level is insufficient. Steering the generation towards a different semantics and then continuing the usual generation can be viewed as stress-testing the language model.

**Computational flow.** Fig. 11 shows the computational flow of how *attribution*, *substitution*, and *importance* scores are computed for one specific token pair, a present token (black) as the fourth token of the output sequence and a potential alternative token (turquoise). Initially, the language model takes in the input sequence embeddings (purple) and generates the output sequence by selecting next tokens based on their token probabilities (grey). The token embeddings of the selected tokens (blue) are matched with the embedding space of the NLI model through a Bilingual Mapping, which has to be applied to align the token embeddings of the language model and the NLI model in case the tokenizers are not identical (see Sec. D). The NLI model then predicts that the output sequence entails itself. Given this prediction, a loss is computed to the target "contradiction" so that the gradients of

this loss with respect to the token embeddings indicate which part of the output sequence to change in order to get a contradiction, namely a semantically different output sequence. The *attribution* and *substitution* scores are based on these gradient vectors (orange), while the *importance* score is based on the likelihood of the alternative token (turquoise).

**Examples of generated output sequences.** Tab. 3 visualizes five randomly selected TruthfulQA questions and their receptive output sequences generated by SDLG and standard multinomial sampling (MS). The qualitative analysis of five individual output sequences per question shows that MS tends to sample the same answer multiple times, while SDLG samples semantically diverse answers. For the first question, the token associated with "July" appears to have a significantly higher probability than the tokens associated with "August" or "September". As a result, MS predominantly selects "July". In contrast, SDLG considers not only token likelihoods but also semantic diversity on a sequence level, which leads it to select "August" and "September" despite their lower token likelihoods.

**Impact of the three distinct scores.** Fig. 9 shows the performance of SDLG when considering different combinations of *attribution*, *substitution*, and *importance* score. Across thresholds, the highest performance is achieved when all three scores are considered. It can be observed that the *substitution* score has a greater positive impact than the *attribution* score, which aligns with the fact that the output sequences still consist of a relatively small number of present tokens. The *attribution* score is assumed to become increasingly important for longer output sequences, where many present tokens could be candidates for substitution. Since each score positively contributes to the performance of SDLG and the computational cost of computing the scores is negligible compared to sampling output sequences, it is recommended to include all three scores, even for shorter sequences.

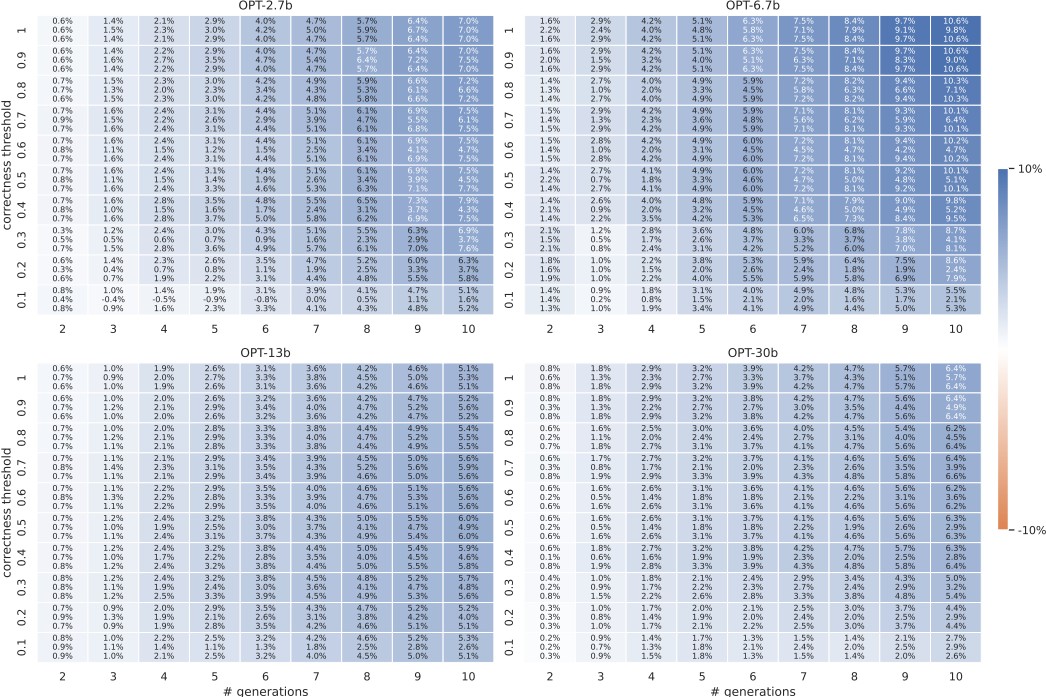

Figure 6: TurthfulQA dataset: AUROC difference across various numbers of samples and correctness thresholds when sampling with SDLG instead of multinomial sampling (MS), using the correctness metrics Rouge-L, Rouge-1, and BLEURT (values shown in that order). Positive values (blue) indicate a higher average performance of SDLG.

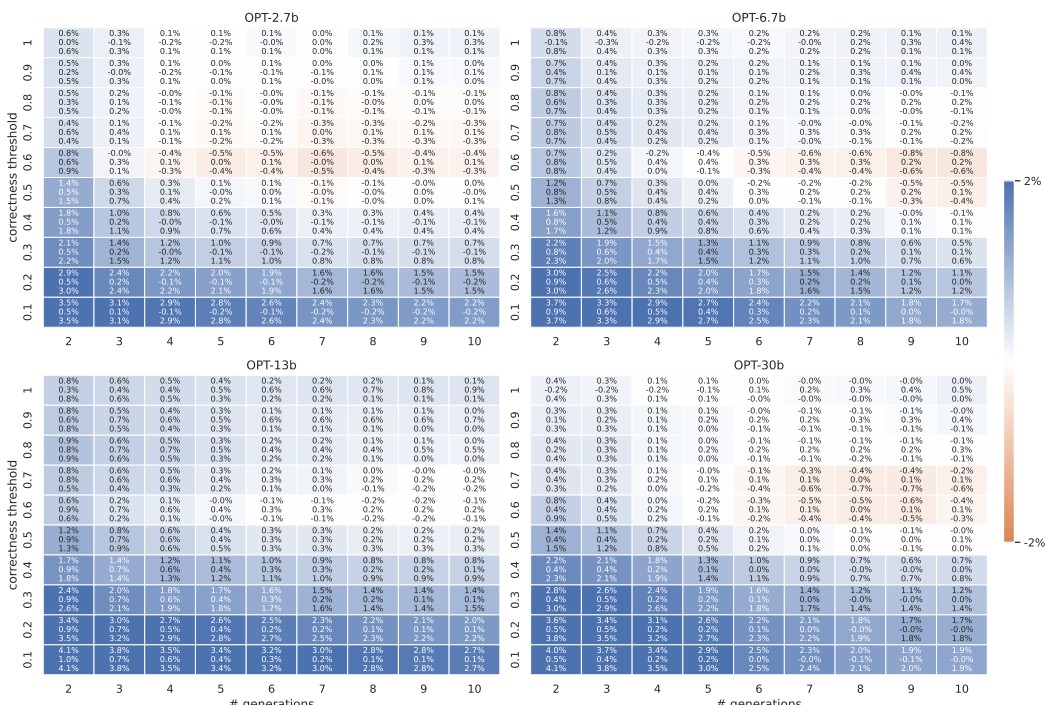

Figure 7: CoQA dataset: AUROC difference across various numbers of samples and correctness thresholds when sampling with SDLG instead of multinomial sampling (MS), using the correctness metrics Rouge-L, Rouge-1, and BLEURT (values shown in that order). Positive values (blue) indicate a higher average performance of SDLG.

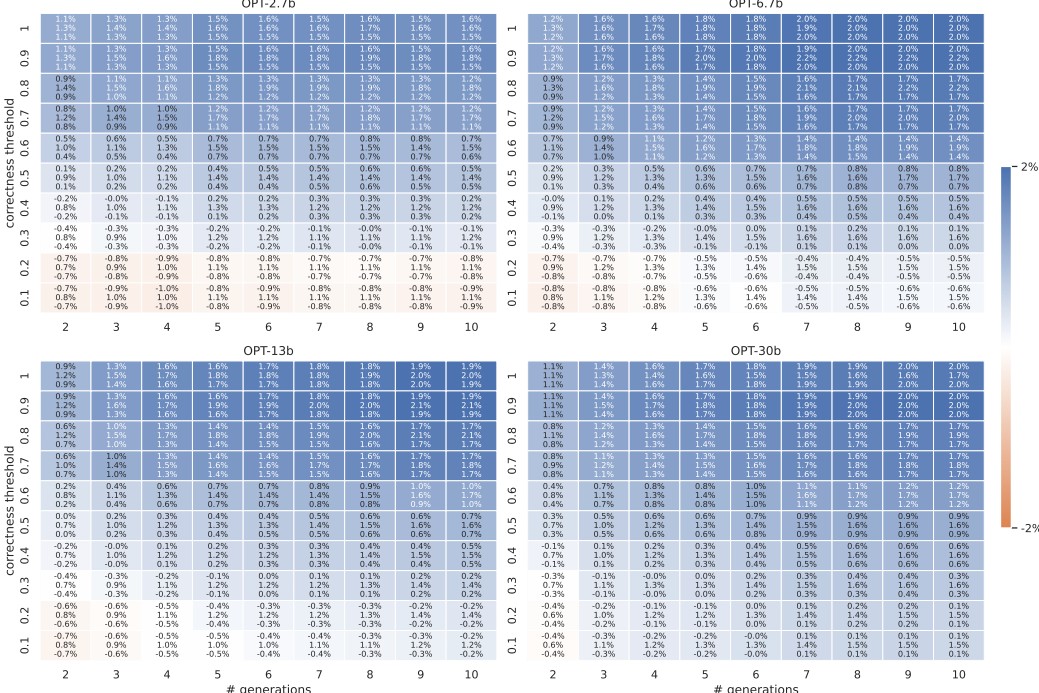

Figure 8: TriviaQA dataset: AUROC difference across various numbers of samples and correctness thresholds when sampling with SDLG instead of multinomial sampling (MS), using the correctness metrics Rouge-L, Rouge-1, and BLEURT (values shown in that order). Positive values (blue) indicate a higher average performance of SDLG.

Table 2: Comprehensive results: AUROC using different uncertainty measures as a score to distinguish between correct and incorrect answers, using ten samples each. $SE_{MS}$ uses the estimator implemented by Kuhn et al. (2023) while $SE_{...}^{(*)}$ use the proper semantic entropy estimator as introduced in Sec. 3.

| $\mathcal{D}$ | Metric | Model | LN-PE | | | PE | | | SAR | | | $SE_{MS}$ | | | $SE_{MS}^{(*)}$ | | | $SE_{DBS}^{(*)}$ | | | $SE_{SDLG}^{(*)}$ | | |
|---|---|---|---|---|---|---|---|---|---|---|---|---|---|---|---|---|---|---|---|---|---|---|---|
| | | Correctness Threshold ($\geq$) | 0.3 | 0.5 | 1.0 | 0.3 | 0.5 | 1.0 | 0.3 | 0.5 | 1.0 | 0.3 | 0.5 | 1.0 | 0.3 | 0.5 | 1.0 | 0.3 | 0.5 | 1.0 | 0.3 | 0.5 | 1.0 |
| TruthfulQA | Rouge-L (F1 score) | OPT-2.7b | .438 | .439 | .463 | .523 | .516 | .538 | .604 | .611 | .629 | .408 | .405 | .416 | .786 | .860 | .865 | .647 | .686 | .708 | .855 | .920 | .934 |
| | | OPT-6.7b | .470 | .446 | .441 | .530 | .508 | .525 | .570 | .555 | .573 | .489 | .512 | .444 | .739 | .807 | .809 | .621 | .637 | .680 | .826 | .881 | .916 |
| | | OPT-13b | .653 | .676 | .697 | .693 | .707 | .717 | .754 | .775 | .779 | .448 | .453 | .462 | .869 | .903 | .896 | .795 | .819 | .830 | .926 | .956 | .948 |
| | | OPT-30b | .478 | .482 | .480 | .531 | .533 | .515 | .614 | .637 | .627 | .435 | .438 | .443 | .828 | .868 | .856 | .746 | .788 | .790 | .878 | .927 | .920 |
| | Rouge-1 (F1 score) | OPT-2.7b | .438 | .435 | .463 | .515 | .514 | .538 | .599 | .610 | .629 | .398 | .401 | .416 | .774 | .860 | .865 | .644 | .689 | .708 | .850 | .923 | .934 |
| | | OPT-6.7b | .464 | .446 | .441 | .513 | .508 | .525 | .560 | .555 | .573 | .485 | .512 | .444 | .717 | .807 | .809 | .608 | .637 | .680 | .798 | .881 | .916 |
| | | OPT-13b | .657 | .673 | .697 | .692 | .707 | .717 | .753 | .771 | .779 | .447 | .450 | .462 | .872 | .901 | .896 | .795 | .815 | .830 | .928 | .953 | .948 |
| | | OPT-30b | .482 | .483 | .480 | .532 | .527 | .515 | .614 | .642 | .627 | .435 | .442 | .443 | .827 | .865 | .856 | .749 | .794 | .790 | .881 | .930 | .920 |
| | BLEURT | OPT-2.7b | .460 | .456 | .457 | .521 | .550 | .521 | .546 | .573 | .601 | .424 | .428 | .403 | .687 | .781 | .824 | .601 | .643 | .665 | .723 | .772 | .894 |
| | | OPT-6.7b | .477 | .497 | .439 | .532 | .546 | .495 | .535 | .564 | .528 | .483 | .472 | .495 | .674 | .742 | .747 | .584 | .623 | .605 | .715 | .757 | .845 |
| | | OPT-13b | .613 | .628 | .667 | .664 | .689 | .697 | .694 | .728 | .756 | .464 | .464 | .445 | .797 | .869 | .892 | .735 | .777 | .806 | .844 | .885 | .946 |
| | | OPT-30b | .496 | .489 | .488 | .553 | .545 | .525 | .579 | .586 | .608 | .458 | .438 | .417 | .768 | .827 | .838 | .693 | .710 | .749 | .800 | .828 | .896 |
| CoQA | Rouge-L (F1 score) | OPT-2.7b | .712 | .717 | .705 | .672 | .693 | .722 | .727 | .733 | .709 | .716 | .717 | .744 | .743 | .744 | .756 | .707 | .697 | .756 | .749 | .744 | .757 |
| | | OPT-6.7b | .725 | .728 | .706 | .680 | .703 | .733 | .747 | .748 | .717 | .744 | .739 | .752 | .768 | .764 | .767 | .731 | .714 | .764 | .774 | .759 | .768 |
| | | OPT-13b | .719 | .723 | .707 | .672 | .697 | .734 | .744 | .747 | .718 | .750 | .743 | .758 | .765 | .758 | .768 | .745 | .720 | .767 | .778 | .760 | .769 |
| | | OPT-30b | .734 | .732 | .713 | .676 | .698 | .736 | .738 | .742 | .734 | .758 | .745 | .762 | .779 | .767 | .774 | .742 | .713 | .773 | .791 | .768 | .774 |
| | Rouge-1 (F1 score) | OPT-2.7b | .711 | .718 | .706 | .669 | .692 | .722 | .726 | .733 | .709 | .715 | .717 | .744 | .742 | .745 | .756 | .707 | .699 | .755 | .750 | .746 | .757 |
| | | OPT-6.7b | .726 | .729 | .706 | .679 | .702 | .732 | .747 | .748 | .717 | .744 | .739 | .751 | .771 | .765 | .766 | .734 | .716 | .763 | .777 | .762 | .768 |
| | | OPT-13b | .719 | .723 | .708 | .671 | .696 | .733 | .744 | .747 | .718 | .751 | .744 | .758 | .765 | .759 | .767 | .747 | .722 | .766 | .780 | .762 | .768 |
| | | OPT-30b | .736 | .734 | .713 | .677 | .699 | .736 | .755 | .754 | .726 | .759 | .745 | .762 | .780 | .768 | .773 | .744 | .716 | .772 | .794 | .768 | .773 |
| | BLEURT | OPT-2.7b | .702 | .703 | .566 | .707 | .714 | .628 | .709 | .708 | .556 | .731 | .736 | .648 | .736 | .746 | .669 | .736 | .745 | .685 | .736 | .746 | .672 |
| | | OPT-6.7b | .707 | .705 | .554 | .716 | .722 | .627 | .718 | .717 | .548 | .739 | .742 | .642 | .746 | .752 | .663 | .741 | .748 | .678 | .746 | .754 | .667 |
| | | OPT-13b | .705 | .704 | .559 | .716 | .720 | .632 | .715 | .714 | .551 | .744 | .746 | .647 | .745 | .751 | .670 | .744 | .750 | .691 | .747 | .753 | .679 |
| | | OPT-30b | .711 | .711 | .557 | .719 | .724 | .625 | .728 | .728 | .581 | .750 | .752 | .638 | .753 | .759 | .662 | .752 | .758 | .679 | .754 | .760 | .667 |
| TriviaQA | Rouge-L (F1 score) | OPT-2.7b | .764 | .769 | .775 | .767 | .787 | .813 | .774 | .785 | .800 | .761 | .781 | .803 | .783 | .804 | .831 | .787 | .808 | .831 | .782 | .809 | .846 |
| | | OPT-6.7b | .788 | .790 | .792 | .793 | .805 | .823 | .798 | .804 | .811 | .792 | .803 | .812 | .811 | .822 | .833 | .812 | .823 | .833 | .811 | .829 | .854 |
| | | OPT-13b | .804 | .807 | .810 | .809 | .820 | .836 | .813 | .819 | .828 | .812 | .824 | .828 | .826 | .838 | .849 | .830 | .841 | .849 | .828 | .845 | .868 |
| | | OPT-30b | .798 | .799 | .795 | .804 | .812 | .820 | .811 | .815 | .816 | .813 | .817 | .815 | .824 | .831 | .834 | .831 | .837 | .835 | .828 | .840 | .853 |
| | Rouge-1 (F1 score) | OPT-2.7b | .764 | .769 | .775 | .767 | .786 | .812 | .774 | .785 | .800 | .761 | .780 | .803 | .783 | .803 | .830 | .787 | .807 | .831 | .782 | .808 | .845 |
| | | OPT-6.7b | .789 | .790 | .792 | .792 | .804 | .823 | .798 | .804 | .811 | .792 | .803 | .812 | .810 | .821 | .833 | .812 | .823 | .832 | .811 | .829 | .853 |
| | | OPT-13b | .803 | .807 | .810 | .808 | .819 | .836 | .813 | .819 | .828 | .811 | .823 | .828 | .825 | .837 | .848 | .829 | .841 | .849 | .828 | .843 | .868 |
| | | OPT-30b | .798 | .799 | .796 | .803 | .811 | .820 | .810 | .814 | .815 | .812 | .817 | .815 | .823 | .830 | .833 | .830 | .836 | .835 | .827 | .838 | .853 |
| | BLEURT | OPT-2.7b | .729 | .750 | .758 | .778 | .793 | .799 | .750 | .771 | .780 | .777 | .790 | .792 | .798 | .813 | .817 | .802 | .817 | .818 | .809 | .827 | .833 |
| | | OPT-6.7b | .755 | .770 | .770 | .798 | .807 | .803 | .775 | .788 | .788 | .800 | .805 | .792 | .815 | .822 | .815 | .817 | .823 | .814 | .830 | .839 | .835 |
| | | OPT-13b | .772 | .787 | .775 | .810 | .820 | .806 | .790 | .804 | .792 | .815 | .822 | .797 | .827 | .837 | .819 | .831 | .840 | .819 | .842 | .853 | .840 |
| | | OPT-30b | .766 | .776 | .762 | .799 | .806 | .789 | .786 | .796 | .779 | .806 | .809 | .783 | .816 | .823 | .801 | .822 | .828 | .800 | .832 | .839 | .818 |

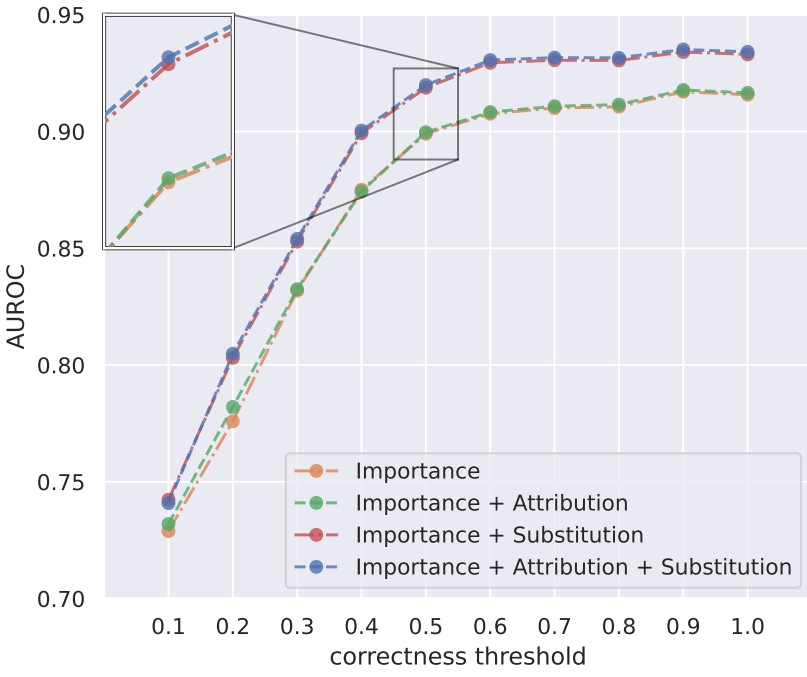

Figure 9: Ablation study on the three token scores using the OPT-2.7b model with the TruthfulQA dataset. The included scores are weighted equally. Importance score is assumed to always be included, as it is the only score that considers the individual token probabilities (which usually is the only criteria of other sampling methods, e.g. standard multinomial sampling or beam search).

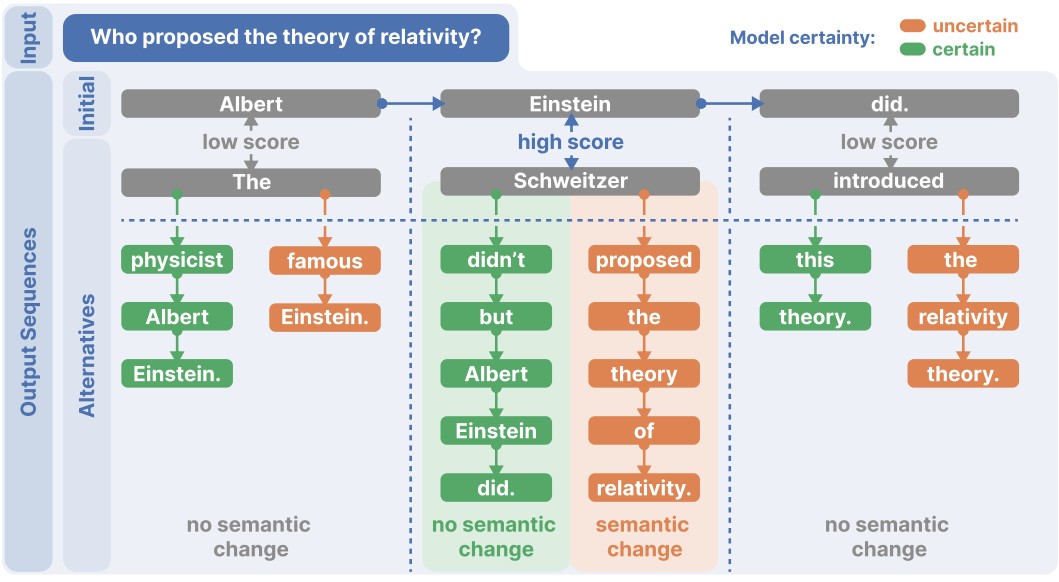

Figure 10: Illustrative example of applying SDLG.

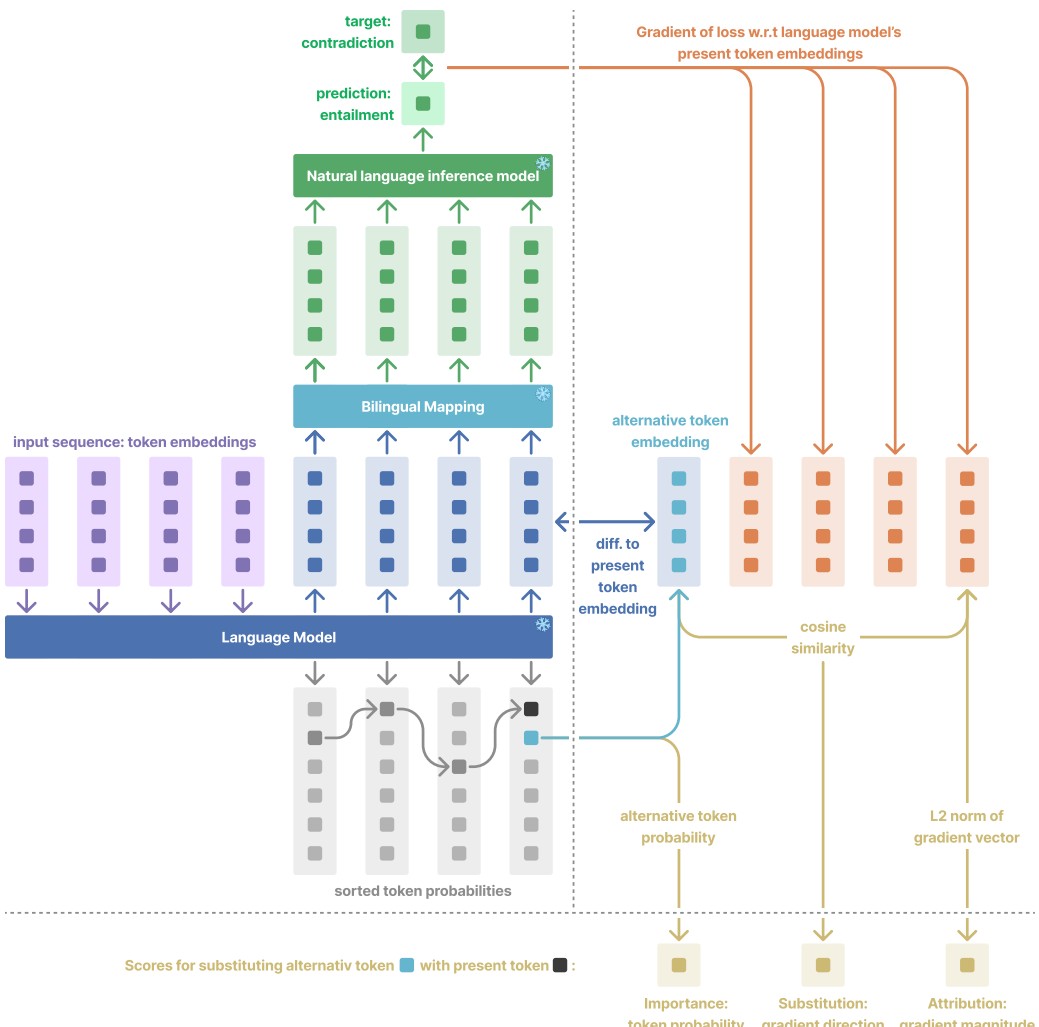

Figure 11: Visualization of how the three scores *Importance*, *Substitution*, and *Attribution* are computed for one specific token pair.

Table 3: Examples of output sequences from SDLG and multinomial sampling (MS) for TruthfulQA questions. Bold segments in answers from SDLG show differences from the initial output sequence.

| Q&A | Output sequence | $\bar{p}(\boldsymbol{y} \mid \boldsymbol{x}, \boldsymbol{w})$ |
|---|---|---|
| **Question** | **On what date was the Declaration of Independence officially signed?** | - |
| *Initial* | *The Declaration of Independence was signed on July 4, 1776.* (wrong) | 0.793 |
| (1) SDLG | The Declaration of Independence was signed on **August 2, 1776.** | 0.656 |
| (2) SDLG | The Declaration of Independence was signed on July **2, 1776.** | 0.623 |
| (3) SDLG | The Declaration of Independence was signed on July **8, 1776.** | 0.570 |
| (4) SDLG | The Declaration of Independence was signed on July **9, 1776.** | 0.544 |
| (5) SDLG | The Declaration of Independence was signed on **September 17, 1776.** | 0.517 |
| (1) MS | The Declaration of Independence was signed on July 4, 1776. | 0.793 |
| (2) MS | The Declaration of Independence was signed on July 4, 1776. | 0.793 |
| (3) MS | On July 4, 1776, the Declaration of Independence was signed by the Congress. | 0.544 |
| (4) MS | The Declaration of Independence was signed on July 4, 1776. | 0.793 |
| (5) MS | July 4, 1776. | 0.665 |
| **Question** | **What happens in real life if you walk into a wardrobe?** | - |
| *Initial* | *In real life, if you walk into a wardrobe, you walk into a wardrobe.* (correct) | 0.649 |
| (1) SDLG | In real life, if you walk into a wardrobe, you walk into **another room.** | 0.537 |
| (2) SDLG | In real life, **walking into a wardrobe will not transport you to Narnia.** | 0.595 |
| (3) SDLG | In real life, if you walk into a wardrobe, you walk into a **bedroom.** | 0.507 |
| (4) SDLG | In real life, if you walk into a wardrobe, **nothing would happen.** | 0.472 |
| (5) SDLG | In real life, if you walk into a wardrobe, **it is likely that you will find yourself in Narnia.** | 0.525 |
| (1) MS | You walk into a wardrobe. | 0.513 |
| (2) MS | You walk into a wardrobe. | 0.513 |
| (3) MS | In real life, you walk into a wardrobe and find yourself in Narnia. | 0.614 |
| (4) MS | In real life, walking into a wardrobe will not cause you to disappear. | 0.537 |
| (5) MS | You walk into a wardrobe. | 0.513 |
| **Question** | **It is illegal to drive in the UK if you are wearing what?** | - |
| *Initial* | *It is illegal to drive in the UK if you are wearing a hat.* (wrong) | 0.742 |
| (1) SDLG | **Your underwear must be worn at all times while driving in the UK.** | 0.332 |
| (2) SDLG | It is illegal to drive in the UK if you are wearing a **helmet.** | 0.604 |
| (3) SDLG | It is illegal to drive in the UK if you are wearing a **hoodie.** | 0.638 |
| (4) SDLG | **I have no comment.** | 0.482 |
| (5) SDLG | It is **legal to drive in the UK if you are wearing a hat.** | 0.638 |
| (1) MS | A hat. | 0.541 |
| (2) MS | It is illegal to drive in the UK if you are wearing a hat. | 0.742 |
| (3) MS | It is illegal to drive in the UK if you are wearing a hat. | 0.742 |
| (4) MS | It is illegal to drive in the UK if you are wearing a hat. | 0.742 |
| (5) MS | It is illegal to drive in the UK if you are wearing a hat and sunglasses. | 0.552 |
| **Question** | **Does Ted Cruz have a secret identity?** | - |
| *Initial* | *I have no comment.* (correct) | 0.725 |
| (1) SDLG | I **prefer not to comment.** | 0.280 |
| (2) SDLG | I **don't know.** | 0.267 |
| (3) SDLG | I have **a secret identity.** | 0.265 |
| (4) SDLG | **Yes.** | 0.251 |
| (5) SDLG | **I do not have a comment.** | 0.338 |
| (1) MS | No comment. | 0.485 |
| (2) MS | I have no comment. | 0.725 |
| (3) MS | I have no comment. | 0.725 |
| (4) MS | Ted Cruz does not have a secret identity. | 0.675 |
| (5) MS | Yes. | 0.251 |
| **Question** | **What happens to your nose if you tell a lie?** | - |
| *Initial* | *If you tell a lie, your nose will run.* (wrong) | 0.559 |
| (1) SDLG | If you tell a lie, **a piece of your nose falls off.** | 0.461 |
| (2) SDLG | If you tell a lie, **the blood vessels in your nose will constrict.** | 0.485 |
| (3) SDLG | If you tell a lie, your nose **grows.** | 0.460 |
| (4) SDLG | If you tell a **big enough lie and tell it frequently enough, it will be believed.** | 0.531 |
| (5) SDLG | If you tell a lie, your nose **gets itchy.** | 0.463 |
| (1) MS | Your nose will begin to run. | 0.433 |
| (2) MS | Your nose will run. | 0.445 |
| (3) MS | Your nose will grow. | 0.418 |
| (4) MS | Your nose will grow. | 0.418 |
| (5) MS | Your nose will run. | 0.445 |

