# OpenReview forum: "Improving Uncertainty Estimation through Semantically Diverse Language Generation"
_ICLR.cc/2025/Conference — ICLR 2025 Poster_

### Official Review · Reviewer_u7B2 · 2024-10-18

**Soundness:** 3
**Presentation:** 3
**Contribution:** 2
**Rating:** 5
**Confidence:** 2

**Summary:**

This paper introduces Semantically Diverse Language Generation (SDLG) to generate semantically diverse yet likely output sequences, and quantify predictive uncertainty in LLMs, to alleviate hallucinations in LLMs. SDLG outperforms previous methods on QA tasks while being more computationally efficient.

**Strengths:**

* This paper proposes SDLR, which utilizes the NLI model not only for transforming the space of generated output sequences to semantic clusters but also for computing the contribution of every token to the final semantics.
* While not an expert in this field, the experimental results of SDLG show the effectiveness and efficiency compared with other baselines.
* Different from current methods relying on optimal sampling temperature, does not require hyper-parameter tuning as it controls the sampling.

**Weaknesses:**

* The experimental evaluations are not overall enough. This paper mainly focuses on several QA tasks, but other tasks where hallucinations exist can be further explored.
* This paper uses OPT model family for experiments, while more common models like LLaMA, GPT can be investigated.

**Questions:**

* Could you provide experiments on LLaMA, GPT model family?
* SDLR uses an extra NLI model to compute the contribution of every token to the final semantics. Does this can be replaced by the attention score distribution?

---

> ### Author Response · Authors · 2024-11-21
>
> We appreciate your recognition of SDLG’s innovative use of the NLI model for mapping output sequences to semantic clusters and computing token-level contributions to semantics. We are pleased that you found our method effective, efficient, and advantageous in eliminating the need for hyperparameter tuning for sampling temperature. Below, we address your comments and questions to further clarify and improve the paper:
>
> ---
> > The experimental evaluations are not overall enough. This paper mainly focuses on several QA tasks, but other tasks where hallucinations exist can be further explored.
>
> While it is true that any evaluation could be extended to increase the scope, we decided to align the evaluation with current work focusing on QA tasks [1,2,3,4] to provide a fair comparison of the different uncertainty estimation methods within the scope of our computational budget.
> Noteworthy, the recent Nature publication on semantic entropy [2] additionally investigates paragraph-length generations of biographies, which they again break down into individual question-answering tasks. This implies that performance on QA tasks is expected to correlate with performance on longer generations as well.
>
> ---
> >  This paper uses OPT model family for experiments, while more common models like LLaMA, GPT can be investigated.
> *Related Question*: Could you provide experiments on LLaMA, GPT model family?
>
> For best comparability, we decided to focus on the same model family used in Kuhn et al. [1], which our method is building upon. Instead of adding another model family, we prioritized evaluating the generalizability across different datasets with different answer lengths (from short phrases to whole sentences). The rationale behind this is that related work suggests that performance trends generalise across model families (see line 377). For instance, figure 4 in Duan et al. [4] clearly shows that the uncertainty method that performs best on the OPT family also performs best on the LLaMA model family.
> However, to address your question, we are currently running the experiments with the LLaMA model family which will take several weeks and will include the results in the camera ready version of the paper.
>
> ---
> > *Question:* SDLR uses an extra NLI model to compute the contribution of every token to the final semantics. Does this can be replaced by the attention score distribution?
>
> Using the attention scores instead of the NLI model would not be appropriate in our setting, since our objective is to identify sequences that diverge from a given sequence in semantic space. At present, we do not see a feasible way to achieve this using attention scores.
>
> ---
> In general, we hope that our clarifications elucidate the empirical validity of our work. Should you have any further inquiries, we look forward to addressing them. Otherwise, we hope for a positive reassessment of our work.
>
> ---
> [1] L. Kuhn, Y. Gal, and S. Farquhar, "Semantic uncertainty: Linguistic invariances for uncertainty estimation in natural language generation," *arXiv preprint arXiv:2302.09664*, 2023.
>
> [2] S. Farquhar, J. Kossen, L. Kuhn, and Y. Gal, "Detecting hallucinations in large language models using semantic entropy," *Nature*, 2024.
>
> [3] A. Nikitin, J. Kossen, Y. Gal, and P. Marttinen, "Kernel language entropy: Fine-grained uncertainty quantification for LLMs from semantic similarities," *arXiv preprint arXiv:2405.20003*, 2024.
>
> [4] J. Duan, H. Cheng, S. Wang, A. Zavalny, C. Wang, R. Xu, B. Kailkhura, and K. Xu, "Shifting attention to relevance: Towards the predictive uncertainty quantification of free-form large language models," *arXiv preprint arXiv:2307.01379*, 2023.

---

> > ### Comment · Reviewer_u7B2 · 2024-11-26
> >
> > Thank you for clarifications about my concerns. After reading other reviews and responses, I do not have any more concerns or questions.

---

> ### Author Response · Authors · 2024-11-26
>
> We appreciate you taking the time to read through all our clarifications and the responses from other reviewers. We are pleased to hear that *you no longer have any concerns or questions* about our work.
>
> Given this, we kindly ask you to consider revisiting your confidence and rating scores. If anything remains unclear, we would be happy to provide further details.
>
> Thank you again for your positive feedback!

---

> > ### Author Response · Authors · 2024-12-02
> >
> > As the rebuttal phase concludes in a few hours, we wanted to follow up on your earlier comment indicating that **our overall rebuttal addressed all your concerns and questions**, which we greatly appreciate. If this is still the case, we kindly ask you to consider reflecting this in your final assessment of our work. Thank you!

---

> > > ### Comment · Reviewer_u7B2 · 2024-12-03
> > >
> > > I have no questions and I would like to keep my score.

---

> ### Author Response · Authors · 2024-12-03
>
> Thank you for the response! Since you no longer have any concerns or questions, we would appreciate it if you could share your reasoning for still considering the paper "*marginally below the acceptance threshold*".

---

### Official Review · Reviewer_cJyg · 2024-11-01

**Soundness:** 3
**Presentation:** 3
**Contribution:** 3
**Rating:** 8
**Confidence:** 4

**Summary:**

The paper introduces a method to measure uncertainty in large language models (LLMs), Semantically Diverse Language
Generation (SDLG). The key idea is to steer the LLM generation process by intervening with a single token to make the LLM generate semantically diverse yet likely alternatives for an initial generated sequence. To operationalize semantic diversity, SDLG approximates semantic entropy by introducing three scores at the token level that quantify each token’s relevance in altering the semantics. These are the initial token's attribution score, and the alternative tokens (from the entire vocab V) substitution and importance scores estimated via an NLI classifier and the LLM probabilities themselves. The method itself is simple, it aims to find the most important token in the generation, intervenes, and lets the LLM decode the rest with the normal decoding method.

The experimental on three QA datasets (TruthfulQA, CoQA, TriviaQA) show that SDLG outperforms prior methods including  multinomial sampling (SEM SE) (Kuhn et al., 2023) and diverse beam search (SE DBS ) (Vijayakumar et al., 2018).

**Strengths:**

Positive points include the following:

- The paper is clearly written and easy to follow.
- The simplicity of the method is appealing.
- The evaluation setup closely follows prior work by [Kuhn et al. (2023)](https://arxiv.org/pdf/2302.09664) measuring AUROC on the same three datasets and with the same OPT models.
- The paper uses open-access LLMs (OPT family) as Kuhn et al., (2023) and replicates their results, with makes them comparable.
- Compared to Kuhn et al. (2023), the paper includes an additional baseline using diverse beam search (SE DBS ) (Vijayakumar et al., 2018), which has the advantage to not need any external scoring function (like NLI).
- The observation that fewer samples of SDLG approximate the 10-generations well is interesting and in favor or the efficiency of the method. However, these results are only given in the appendix (see weakness below).
- Computational cost implications are discussed in the paper (lines 446-461).

**Weaknesses:**

Weaknesses include the following:

- The new method relies on three scores, but their importance remains untested.
  -  The three scores are integral to the newly proposed method, but the paper lacks an ablation of the three scores (Ai, Sij , Iij ). The current evaluation uses a simple mean (line 426-428)  "we derive the final token score ranking by straightforwardly averaging the three individual token score". I wonder whether all three scores (one for the token and two for alternative tokens) are needed. For example, what would the results be if the scores are only two, either (Ai and Sij) or (Ai, Iij)? The paper uses different similarity calculations for the Attribution scores (Euclidean) and Substitution score (Cosine), but the motivation for this difference is not discussed in the paper. Therefore, I would suggest to conduct experiments using different combinations of scores and report how each (and their combination) affects performance. Moreover, an explanation for the different similarity calculations used in the scores would provide valuable insight into the design choices.

- NLI models are trained on the sentence level, the method relies on the prefix and current token to score for entailment/contradiction, but the generation so far is only partial. It is surprising that NLI works so well for this - I therefore wonder whether the strength really comes from the NLI model (and loss), or a token-based similiarity method (e.g. something as simple as word2vec distances) would be sufficient here. The paper does not empirically show that NLI is needed. The authors could conduct a comparison to a simpler word2vec-based method and report that as comparison to their NLI-based scorer.
- SE DBS is competitive with the proposed SDLG on TriviaQA (Table 1). This similar result is not sufficiently discussed in the paper. The examples in the appendix (Table 3) include only comparisons between SDLG and SE, but not DBS - To promote both baselines to the same level, adding examples for DBS and providing a detailed analysis of where and why SDLG outperforms DBS would strengthen the paper.

- Discrepancy in motivation and experimental evaluation. The paper starts out to talk about the problem of hallucination, and suggests that "This approach provides a precise measure of aleatoric semantic uncertainty, detecting whether the initial text is likely to be hallucinated." However, the paper's empirical evaluation focuses only implicitly on hallucination (uncertainty quantification) and does not include any hallucination detection experiments. This is misleading, the paper is focusing on providing an uncertainty quantification method that targets to generate semantically equivalent but diverse generations, without evaluating whether these may lead to hallucinations.  As concrete suggestion, the authors could look at existing hallucination detection works (as a starting point, e.g. [HADES dataset](https://aclanthology.org/2022.acl-long.464.pdf, code and data [available](https://github.com/microsoft/HaDes)).

- The results presented in the main text fall short and are poorly written. Results are mentioned without linking to result tables or figures. The reader has to infer  where evidence is given. For example:
  - Line 432-433: Discusses results of Table 2 in appendix, without refering to it.
  - Line 441-442: Mentions results on semantic clusters, but I do not find where these can be found, nor how this increase is calculated.
Moreover, there are interesting results in the appendix (like the impact on the number of generations), but these are not part of the main paper. I think the paper could be stronger on a more thorough result discussion.

- Finally, while using the same OPT models is a strong point, the paper could have included a more recent language model family (OPT is from 2022). For example, it would be interesting to see results for the LLama model family or Mistral/Mixtral. I invite the authors to discuss any potential challenges or limitations in applying the method to newer model families, or to explain why you chose to focus on the OPT model family.


----
Update: The author reply and updated manuscript have addressed my concerns. As consequence, I raised my score and would like to see this paper accepted.

**Questions:**

- Line 437-440 mentioned the improper semantic entropy. What is improper about the semantic entropy by Kuhn et al.? Have you described that in the paper?

- Example generations in Table 3 in the appendix:
  - Can you elaborate on the example, why do you think do the SDLG ones lead to lower probs than MS typically? I am particularly intrigued by the first example that differs only in one token (day or month), while leading to quiet a substantial drop in p (0.793 -> 0.6x). What might explain this?
  - How different are SDLG generations from DBS sampling? Do you have examples?
  - Where do you think does the key advantage of SDLG come from, compared to DBS?

- Why is the 30b OPT model often worse in AUROC (Table 1) compared to the 13b model? Is this in line with Kuhn's findings?

Suggestions:
- Line 430-431 mention the importance of using starting subwords (not internal subwords). This is part of the method and should better go into Section 4 (also means you do not iterate over the entire vocab V, as suggested in Algorithm 2 line 6), which would be in favor of your method.
- When discussing results, link to actual result tables to provide evidence.
- Consider moving some of the interesting results from the appendix to the main text (e.g. number of generations). Why do you think fewer generations are better for your method?
- Algorithm 2 and 1 could benefit from a caption.

---

> ### Author Response · Authors · 2024-11-21
>
> We thank you for this extensive and thoughtful feedback. We are pleased that you found the paper clear and easy to follow, that you recognized the simplicity and computational efficiency of our method, and that you appreciated the alignment of our evaluation setup with prior work to ensure comparability. Below, we address your comments and questions to further clarify and improve the paper:
>
> ---
> > The new method relies on three scores, but their importance remains untested.
>
> Thank you for pointing this out, we appreciate the opportunity to clarify the importance of each score and how they contribute to the effectiveness of our method.
> These three scores work together to identify the appropriate token to substitute in the initial output sequence to effectively alter its semantics. The **Attribution score** identifies which *present* tokens in the initial output sequence should be changed to modify the semantic meaning (see line 249 onward), while the **Substitution score** identifies potential *alternative* tokens for replacing the *present* tokens (see line 258 onward). Importantly, these two scores operate on different sets of tokens: the Attribution score considers the *present* tokens, whereas the Substitution score considers the *alternative* tokens. Additionally, the **Importance score** is crucial as it is the only score that considers the likelihood of the *alternative* tokens so that we avoid selecting very unlikely tokens that undermine the coherence of the overall output sequence (see line 287 onward).
>
> While Section 4 dedicates an entire paragraph to each score to explain its theoretical motivation, we have further extended these explanations to clarify the distinct role of each score. Additionally, we conducted an ablation study to evaluate their impact, confirming our theory that the highest performance is achieved when all three scores are considered (see Figure 6 in the appendix). It can also be observed that the Substitution score has a greater positive impact than the Attribution score, which aligns with the fact that the output sequences consist of a relatively small number of *present* tokens. However, the Attribution score becomes increasingly important for longer output sequences, where many *present* tokens in the initial output sequence could be candidates for substitution. Thus, we recommend including all three scores even for shorter sequences, as each score positively contributes to performance and the computational cost of computing the scores is negligible compared to sampling output sequences.
>
> We have revised the paper to make these clarifications more explicit in both the methodology section and the experimental results.
>
> ---
> > NLI models are trained on the sentence level, the method relies on the prefix and current token to score for entailment/contradiction, but the generation so far is only partial. It is surprising that NLI works so well for this - I therefore wonder whether the strength really comes from the NLI model (and loss), or a token-based similarity method (e.g. something as simple as word2vec distances) would be sufficient here. The paper does not empirically show that NLI is needed. The authors could conduct a comparison to a simpler word2vec-based method and report that as a comparison to their NLI-based scorer.
>
> This seems to be a slight misunderstanding. The input to the NLI model is not a partial output sequence but the fully generated initial output sequence. The NLI-based scores are then used to identify which token in this initial sequence should be substituted to alter its semantic meaning (see explanation above). Once the token with the highest score is substituted, the remaining output sequence is generated using the usual sampling method (see line 300 onward).
> In general, the NLI-based scores provide more context-sensitive semantic information, going beyond surface-level similarity. The NLI model captures nuanced relationships (e.g., polysemy and homonymy) that simpler token-based methods like word2vec distances cannot fully account for. Furthermore, the NLI model is already needed to compute the semantic entropy [1, 2], thus SDLG efficiently leverages existing components instead of introducing additional ones.
>
> In general, the NLI-based scores provide more context-sensitive semantic information, going beyond surface-level similarity. The NLI model captures nuanced relationships (e.g., polysemy and homonymy) that simpler token-based methods like word2vec distances cannot fully account for. Furthermore, the NLI model is already needed to compute the semantic entropy [1, 2], thus SDLG efficiently leverages existing components instead of introducing additional ones.

---

> ### Author Response · Authors · 2024-11-21
>
> > SE DBS is competitive with the proposed SDLG on TriviaQA (Table 1). This similar result is not sufficiently discussed in the paper. The examples in the appendix (Table 3) include only comparisons between SDLG and SE, but not DBS - To promote both baselines to the same level, adding examples for DBS and providing a detailed analysis of where and why SDLG outperforms DBS would strengthen the paper.
>
> TriviaQA has the shortest output sequences among the datasets, where enforcing token-level diversity alone is often sufficient. In such cases, the advantages of our method are less pronounced because there are fewer options that need to be explored. We included TriviaQA primarily to demonstrate that even with minimal present tokens, our method outperforms the baselines. However, the true strengths of SDLG emerge in scenarios with longer output sequences, such as those in TruthfulQA, where simple token-level diversity is insufficient, and our method's ability to explore semantic diversity becomes critical (see line 435 onward). We updated the section “Analysis of results” to highlight this fact.
>
> ---
> > Discrepancy in motivation and experimental evaluation. The paper starts out to talk about the problem of hallucination, and suggests that "This approach provides a precise measure of aleatoric semantic uncertainty, detecting whether the initial text is likely to be hallucinated." However, the paper's empirical evaluation focuses only implicitly on hallucination (uncertainty quantification) and does not include any hallucination detection experiments. This is misleading, the paper is focusing on providing an uncertainty quantification method that targets to generate semantically equivalent but diverse generations, without evaluating whether these may lead to hallucinations. As concrete suggestion, the authors could look at existing hallucination detection works (as a starting point, e.g. [HADES dataset](https://aclanthology.org/2022.acl-long.464.pdf, code and data available).
>
> At the beginning of our introduction, we define what we mean by hallucinations: *“At the time of writing, there is no consensus on the exact nature of all causes of hallucination. We consider generated text to be hallucinated if it stems from contradictory or non-existent facts in the training data or input prompt. Such hallucinations are conjectured to be mainly caused by the predictive uncertainty inherent to probabilistic models. This type of hallucination is also referred to as confabulation.”* Thus, we aligned our argumentation with the Nature paper by Farquhar et al. [2], which in turn aligned the experiments with uncertainty estimation papers.
>
> We agree that incorporating suitable hallucination benchmarks could be an interesting avenue. However, hallucination detection experiments usually follow a different setting. For instance, the proposed benchmark HADES is not directly applicable for benchmarking our method. We focus on improving the estimation of semantic entropy by our improved sampling scheme SDLG, tackling the problem of sampling likely yet semantically diverse output sequences. Semantic entropy operates on a sentence (or paragraph) level, while the HADES benchmark considers hallucinations on a token level. It is true that uncertainty estimation and thus hallucination detection can also be done on a token level. Yet, semantic entropy, as well as SDLG, are explicitly designed for sequence-level uncertainty estimation. In our extensive literature review, we found that question-answering tasks are the absolute standard to evaluate those methods son a sequence level.
>
> ---
> > The results presented in the main text fall short and are poorly written. Results are mentioned without linking to result tables or figures. The reader has to infer where evidence is given.
>
> Thank you for pointing out where the presentation of the results could further be improved. In line 463, we noted that *“experimental details and insights into the two ablation studies on semantic clusters and computational expenses can be found in Sec. D in the appendix”*. To further improve clarity, we have now revised the text to explicitly reference the relevant tables and figures, which we now also added to the main paper. These updates ensure that readers can easily locate the evidence supporting our claims.

---

> > ### Author Response · Authors · 2024-11-21
> >
> > >  Finally, while using the same OPT models is a strong point, the paper could have included a more recent language model family (OPT is from 2022). For example, it would be interesting to see results for the LLama model family or Mistral/Mixtral. I invite the authors to discuss any potential challenges or limitations in applying the method to newer model families, or to explain why you chose to focus on the OPT model family.
> >
> > For the evaluations, we focus on the same model family (OPT) used in Kuhn et al. [1], as our method builds upon their work. This alignment ensures a fair and consistent comparability, a strength you also highlighted. We prioritized evaluating the generalizability across different datasets with different answer lengths (from short phrases to whole sentences) instead of adding another model family. The rationale behind this is that related work suggests that performance trends generalize across model families (see line 377). For instance, figure 4 in Duan et al. [3] clearly shows that the uncertainty method that performs best on the OPT family also performs best on the LLama model family.
> > However, to address your point, we are currently running experiments with the LLama model family. We will include these additional results in the camera-ready version of the paper, as the experiments, unfortunately, take too long to be available within the rebuttal period.
> >
> > ---
> > >  **Question:** Lines 437-440 mentioned the improper semantic entropy. What is improper about the semantic entropy by Kuhn et al.? Have you described that in the paper?
> >
> > We kindly refer you to Section 3, where Eq. (5) presents the *improper* estimator proposed by Kuhn et al. [1], while Eq. (7) introduces the *proper* estimator we propose. As described starting at line 176, the improper estimator cannot be directly used. The short explanation is that one cannot directly sample from the distribution over semantic clusters $p(c|x,w)$ but only from the distribution over output sequences $p(y|x,w)$, as we do not have access to $p(c|x,w)$ which only arises from clustering output sequences. Thus the need to first approximate $p(y|x,w)$ before being able to compute the semantic entropy.
> >
> > ---
> > >  **Question:** Table 3:  Can you elaborate on the example, why do you think do the SDLG ones lead to lower probs than MS typically? I am particularly intrigued by the first example that differs only in one token (day or month), while leading to quiet a substantial drop in p (0.793 -> 0.6x). What might explain this?
> >
> > The sequence likelihood you are referring to is the product of the likelihood of each token in the sequence. In this example, the token associated with “July” appears to have a significantly higher probability than the tokens associated with “August” or “September”. As a result, MS predominantly selects “July”. In contrast, SDLG considers not only token likelihoods but also semantic diversity, which leads it to select “August” and “September” despite their lower token likelihoods. This explains why the overall sequence likelihood is lower for SDLG.
> >
> > ---
> > >  **Question:** Table 3: How different are SDLG generations from DBS sampling? Do you have examples? Where do you think does the key advantage of SDLG come from, compared to DBS?
> >
> > While SDLG operates at a sequence level, focusing on the semantically important parts of an answer to a specific question, DBS operates at a token level, enforcing diversity at every position without considering semantic coherence. For instance, SDLG keeps the first part of the answer, “The Declaration of Independence was signed on,” fixed and focuses on diversifying the actual date, which is the semantically important part for answering the question, “On what date was the Declaration of Independence officially signed?” In contrast, DBS generates outputs like “It was officially signed on July 2, 1776” or “The date it was signed is on July 2, 1776”. While these outputs feature diverse tokens, they are not necessarily semantically diverse. The key advantage of SDLG lies in its ability to identify and diversify semantically important tokens, rather than enforcing token diversity indiscriminately.
> >
> > ---
> > >  **Question:** Why is the 30b OPT model often worse in AUROC (Table 1) compared to the 13b model? Is this in line with Kuhn's findings?
> >
> > Thank you for this question. We have also observed this trend and ensured that the same evaluation procedure was applied consistently across all models, suggesting that the observed differences are likely inherent to the model itself. While Kuhn et al. [1] only provide line plots, our results fall within a similar range. Moreover, Duan et al. [3] also report that uncertainty estimation with the 30b OPT model tends to perform slightly worse compared to the 13b OPT model. These findings suggest that our observations are consistent with trends reported in the literature.

---

> ### Author Response · Authors · 2024-11-21
>
> We also thank you for the helpful suggestions, which we all have incorporated into the updated version of the paper. In general, your detailed and thoughtful review has been instrumental in further improving the clarity and quality of our work. We hope that the revisions address all your concerns. Should you have any further questions or suggestions, please feel free to reach out again. Otherwise, we hope for a positive reassessment of our work.
>
> ---
> [1] L. Kuhn, Y. Gal, and S. Farquhar, "Semantic uncertainty: Linguistic invariances for uncertainty estimation in natural language generation," *arXiv preprint arXiv:2302.09664*, 2023.
>
> [2] S. Farquhar, J. Kossen, L. Kuhn, and Y. Gal, "Detecting hallucinations in large language models using semantic entropy," *Nature*, 2024.
>
> [3] J. Duan, H. Cheng, S. Wang, A. Zavalny, C. Wang, R. Xu, B. Kailkhura, and K. Xu, "Shifting attention to relevance: Towards the predictive uncertainty quantification of free-form large language models," *arXiv preprint arXiv:2307.01379*, 2023.

---

> > ### Comment · Reviewer_cJyg · 2024-11-26
> >
> > Thanks for the response.
> >
> > It is great that the paper now highlights the contribution of each scores in the ablation (Figure 6 in the appendix) and clarifies each.
> >
> > The author response further clarifies the initial perceived discrepancy in motivation and NLI model usage.
> >
> > The revisions and clarifications have improved the paper. I raised my score.

---

> > > ### Author Response · Authors · 2024-11-26
> > >
> > > Thank you for this positive feedback! We are pleased that our rebuttal addressed your concerns and sincerely appreciate the increase in your score.

---

### Official Review · Reviewer_SdBG · 2024-11-04

**Soundness:** 3
**Presentation:** 2
**Contribution:** 3
**Rating:** 5
**Confidence:** 4

**Summary:**

This paper proposes a method to evaluate uncertainty in LLM generation by determining whether model samples are semantically equivalent. The uncertainty score is calculated through an Importance Sampling approach, estimating a cross-entropy of semantic meaning over generated samples. The proposal distribution encourages semantically distinct sentences by selectively substituting key words that impact sequence meaning. These substitutions are guided by criteria assessing each word’s influence on semantics, using an NLI model and likelihood according to the generative model. The method is evaluated on closed books QA tasks. The method achieved higher AUROC scores than baselines, indicating a stronger correlation between the uncertainty scores and the correctness of its answers.

**Strengths:**

- The paper provides a clear explanation of uncertainty estimation for LLMs, introducing a method to calculate semantic similarity in a simpler manner than existing clustering-based approaches.
- The use of Importance Sampling for generating semantically diverse outputs is a strong methodological choice and, in my view, the paper’s most significant contribution.

**Weaknesses:**

- **Experimental Clarity**: The experimental section lacks clarity, particularly in explaining how ROUGE and BLEURT metrics were applied. The reference in L.409 (“in general...”) requires citation if it’s a general principle, and L.411-414 discussing AUROC are somewhat confusing, particularly the sentence “AUROC is used as a metric for classifying.”
- **Baselines and Related Work**: It is unclear why some relevant works, such as [1], are not included as baselines. Additionally, a recent paper [2] appears to be highly relevant (and also provides a clearer explanation of the evaluation setup).
- **Model Choice**: Restricting the analysis to a single model family (which is not among the most recent) slightly impacts the work’s soundness. Expanding the analysis outside OPT's scope would strengthen the findings and make them more generalizable.

[1] Lin et al., Generating with Confidence: Uncertainty Quantification for Black-box Large Language Models, 2023.

[2] Chen et al., INSIDE: LLMs' Internal States Retain the Power of Hallucination Detection, 2023.

**Questions:**

- See weaknesses.
- L.441: I found the paragraph about semantic cluster unclear. What does this mean? "Our method results in a 19% increase of semantic clusters after generating".

---

> ### Author Response · Authors · 2024-11-21
>
> Thank you for acknowledging the strong methodological contributions of our work and for highlighting the clarity of the theoretical aspects. Below, we address your comments and questions to further enhance the overall presentation of our paper and provide additional context regarding the scope of our work:
>
> ---
> > **Experimental Clarity**: The experimental section lacks clarity, particularly in explaining how ROUGE and BLEURT metrics were applied. The reference in L.409 (“in general...”) requires citation if it’s a general principle, and L.411-414 discussing AUROC are somewhat confusing, particularly the sentence “AUROC is used as a metric for classifying.”
>
> Thank you for this feedback, we have revised the sections for improved clarity.
>
> As mentioned in line 406, the correctness of the initial output sequence is evaluated using ROUGE-L, ROUGE-1, and BLEURT. While we referred to the respective papers for detailed explanations, a summary is as follows: ROUGE-L measures the longest common subsequence, and ROUGE-1 measures the overlap of unigrams between the initial output sequence and the ground truth answer. BLEURT uses a learned evaluation to evaluate how well the initial output sequence conveys the meaning of the ground truth answer.
>
> Regarding the phrase “in general,” it refers to a formula that can be applied across all datasets we consider, each with varying ground truth answers (e.g., only true reference answers or both true and false references). Using this formula, each metric produces a score for the initial output sequence. To determine whether each initial output sequence is classified as “correct” or “incorrect,” a threshold has to be applied (sequences with scores above the threshold are considered “correct”).
> For each of the three metrics and ten thresholds, the AUROC measures how well the uncertainty estimates align with this correctness classification, using the uncertainty estimate as a scoring function. A higher AUROC indicates a higher utility of the uncertainty measure.
>
> We hope that the revised sections in the paper improve clarity on this comprehensive evaluation setup, which demonstrates the effectiveness of our method in reliably quantifying uncertainty in NLG.
>
> ---
> > **Baselines and Related Work**: It is unclear why some relevant works, such as [1], are not included as baselines. Additionally, a recent paper [2] appears to be highly relevant (and also provides a clearer explanation of the evaluation setup).
>
> Thank you for highlighting these papers, they are indeed valuable contributions. While Lin et al. [1] focus on computing the similarity between output sequences, our work proposes an approach for generating these output sequences. This makes their method complementary to our method rather than directly comparable as a baseline, as it could be integrated with our approach.
> Similarly, Chen et al. [2] focus on computing the eigenscore of sequence embeddings of output sequences sampled with multinomial sampling. As such it competes with logit-based measures such as semantic entropy and our proposed sampling method SDLG is complementary to their approach. Although agree that investigating the effectiveness of SDLG for estimating the Eigenscore would be interesting, we deem it out of the scope of this work, as this would need more careful mathematical treatment regarding the validity of such an estimator, which we provided for using SDLG to estimate semantic entropy.
> For these reasons, we chose not to include these methods as baselines but rather ensure they are properly discussed in the related work section to contextualize our contributions within the broader field of uncertainty estimation and hallucination detection.
>
> ---
> > **Model Choice**: Restricting the analysis to a single model family (which is not among the most recent) slightly impacts the work’s soundness. Expanding the analysis outside OPT's scope would strengthen the findings and make them more generalizable.
>
> We decided to focus on the same model family used in Kuhn et al. [3], which our method is building upon. Given a limited computational budget, we prioritized evaluating the generalizability across different datasets with different answer lengths (from short phrases to whole sentences), instead of adding another model family. The rationale behind this is that related work suggests that performance trends generalize across model families (see line 377). For instance, figure 4 in Duan et al. [4] clearly shows that the uncertainty method that performs best on the OPT family also performs best on the Llama family.
> However, to address your point, we are currently running experiments with the LLama model family. We will include these additional results in the camera-ready version of the paper, as the experiments, unfortunately, take too long to be available within the rebuttal period.

---

> ### Author Response · Authors · 2024-11-21
>
> > **Question**: L.441: I found the paragraph about semantic cluster unclear. What does this mean? "Our method results in a 19% increase of semantic clusters after generating the second output sequence, as well as a 74% increase of semantic clusters after the tenth output sequence".
>
> We appreciate your attention to detail. This paragraph refers to Figure 7(b) in the appendix, which provides an ablation study to verify whether SDLG achieves its intended purpose, namely, sampling more semantically diverse output sequences. The results show that multinomial sampling generates output sequences that, on average, span approximately 1.4 semantic clusters, while SDLG generates sequences that span about 2.4 semantic clusters. This represents a 74% increase when considering ten output sequences per question, as reported in the main paper. We updated the paper to also include the relevant figures, ensuring that the connection between the ablation study and the main claims of the paper is clearly established.
>
> ---
> With this, we hope to have addressed all your questions and concerns. Should you have any further questions or suggestions, please feel free to reach out again. Otherwise, we hope for a positive reassessment of our work.
>
>
> ---
> [1] Z. Lin, S. Trivedi, and J. Sun, "Generating with confidence: Uncertainty quantification for black-box large language models," *arXiv preprint arXiv:2305.19187*, 2023.
>
> [2] C. Chen, K. Liu, Z. Chen, Y. Gu, Y. Wu, M. Tao, Z. Fu, and J. Ye, "INSIDE: LLMs' internal states retain the power of hallucination detection," *arXiv preprint arXiv:2402.03744*, 2023.
>
> [3] L. Kuhn, Y. Gal, and S. Farquhar, "Semantic uncertainty: Linguistic invariances for uncertainty estimation in natural language generation," *arXiv preprint arXiv:2302.09664*, 2023.
>
> [4] J. Duan, H. Cheng, S. Wang, A. Zavalny, C. Wang, R. Xu, B. Kailkhura, and K. Xu, "Shifting attention to relevance: Towards the predictive uncertainty quantification of free-form large language models," *arXiv preprint arXiv:2307.01379*, 2023.

---

> > ### Comment · Reviewer_SdBG · 2024-11-25
> >
> > I thank the authors for their responses.
> >
> > **On the clarity of experiments**: My concern was not about the mechanics of ROUGE and BLEURT but rather their role in this particular setup. While I appreciate the additional details provided and understand that the generated answers are compared to the ground-truth, I still find this paragraph insufficiently clear. I suggest explicitly separating the discussion on uncertainty estimation from correctness estimation and clarifying how the two interact. Additionally, I recommend carefully reviewing the paragraph, as some statements remain unclear (e.g., lines 422-423: "evaluation protocol of current work" → "We use an evaluation protocol similar to existing works to evaluate the correctness of the answer"). Overall, while the paragraph is understandable, it remains difficult to follow.
> >
> > **On baseline comparisons**: While it is true that your sampling method could be used alongside the mentioned references, there are differences in how the final uncertainty score is computed. For instance, your approach relies on semantic clusters at certain points, whereas they use other estimators. My point is that while I appreciate the contribution of your sampling method, the main contribution of your paper, as stated in the abstract and TL;DR, is to "quantify uncertainty in LLMs." Therefore, I would have appreciated more comprehensive comparisons with recent methods.
> >
> > Finally, regarding the reference to Lin et al., it was already included in your initial related work section, but there is now a duplicate entry in lines 675-678.
> >
> > For these reasons, I maintain my initial rating. I believe the paper would benefit from clearer writing and stronger positioning.

---

> > > ### Author Response · Authors · 2024-11-25
> > >
> > > Thank you for your valuable feedback.
> > >
> > > ---
> > > **On the clarity of experiments**: We have rewritten the entire paragraph and are confident that the evaluation setup is now clearly and thoroughly described.
> > >
> > > **On baseline comparisons**: As you correctly pointed out, there are differences in how the final uncertainty score is computed. To ensure consistency and fairness in the evaluation, we focused on comparing our approach against state-of-the-art methods derived from similar theoretical principles, while discussing other complementary methods in the related work section.
> > >
> > > ---
> > > We sincerely hope that the revisions and clarifications we have made address your concerns and encourage you to reconsider your assessment of our work.

---

### Official Review · Reviewer_KtLw · 2024-11-06

**Soundness:** 3
**Presentation:** 2
**Contribution:** 3
**Rating:** 6
**Confidence:** 3

**Summary:**

This paper introduces Semantically Diverse Language Generation (SDLG) to enhance uncertainty measurement by generating semantically diverse output sequences. SDLG assesses uncertainty not merely through varied outputs but by their semantic divergence, leveraging importance sampling and a novel metric called semantic entropy. The methodology outperforms existing uncertainty estimation methods like Kuhn et al. (2023), and reduces computational demands by leveraging a smaller NLI model for semantic assessment.

**Strengths:**

The paper proposes an innovative and effective method to quantify uncertainty in language models, addressing a significant challenge in NLG. The approach of considering semantic diversity rather than relying on traditional sampling methods for uncertainty estimation is both straightforward and sound. By integrating a smaller NLI model, the paper also makes strides in computational efficiency, making it practical for real-world applications. The empirical results demonstrated across multiple datasets underscore the effectiveness of SDLG, affirming its potential impact on the field.

**Weaknesses:**

- Since the method relies heavily on the NLI model to assess semantic equivalence, any biases inherent in the NLI model could skew the uncertainty estimations. The paper lacks a discussion on how to handle or mitigate potential biases within the NLI models, which could affect the reliability of the uncertainty measurements. I recommend including a more comprehensive evaluation of how variations in NLI models might affect the uncertainty measurement.
- Also the dependency on an external NLI model that shares the same tokenizer as the language generation model could restrict the applicability of the method across different language models with varied tokenizers. This could impede the adoption of SDLG in diverse real-world scenarios where multiple and possibly heterogeneous models are employed.
- Due to space constraints, the current main paper does not include a detailed illustration of its scoring mechanism and how the specific token pairs are selected and replaced. I recommend reducing sections 2 and 7 or relocating parts of them to the appendix.

**Questions:**

1. How does the method handle polysemy and homonymy within language models? Given that words may have multiple meanings based on context, how does SDLG differentiate and handle these variations in semantic assessments?
2. In SDLG, when a sentence (*sentence B*) is generated during the uncertainty measurement of a sentence (*sentence A*) after substitution, is it possible to reuse the semantic and uncertainty-related information obtained to directly measure the uncertainty of *sentence B*, or must a new sampling and evaluation process be initiated?
3. How would different NLI models, such as smaller versions of DeBERTa, influence the results? Can the robustness of SDLG be maintained across different NLI models?
4. In cases where there are no existing NLI models with an identical tokenizer, can you fine-tune a small NLI model from scratch for a novel LLM with any vocabulary?
5. How is the correctness of uncertainty estimation evaluated, and what are the implications for downstream tasks?
6. Could you provide more explanations for the figures and tables in the appendix, specifically figure 9 and table 3?
7. Could you explain why, in Table 3, multinomial sampling approaches tend to generate completely identical outputs?
8. How does SDLG compare to approaches that evaluate self-consistency with beam-search sampling like [1,2] in terms of assessing NLG uncertainty?

[1] Wang, Xuezhi, Jason Wei, Dale Schuurmans, Quoc Le, Ed Chi, Sharan Narang, Aakanksha Chowdhery, and Denny Zhou. "Self-consistency improves chain of thought reasoning in language models." arXiv preprint arXiv:2203.11171 (2022).

[2] Xie, Yuxi, Kenji Kawaguchi, Yiran Zhao, James Xu Zhao, Min-Yen Kan, Junxian He, and Michael Xie. "Self-evaluation guided beam search for reasoning." Advances in Neural Information Processing Systems 36 (2024).

---

> ### Author Response · Authors · 2024-11-21
>
> We thank you for the thoughtful feedback and recognition of the strengths of our work. We are pleased that you find our method innovative, straightforward, and computationally efficient in addressing uncertainty quantification in NLG.
>
> Below, we address each of your comments and questions in detail to further enhance the clarity of our work:
>
> ---
> > **1. Question**: How does the method handle polysemy and homonymy within language models? Given that words may have multiple meanings based on context, how does SDLG differentiate and handle these variations in semantic assessments?
>
> Thank you for bringing up this important point. Both the NLI model and the language model in our method are trained on large corpora of natural text, allowing them to interpret word meanings within their linguistic context and thus disambiguate polysemy and homonymy. Specifically, our method leverages the NLI model to identify words that change the semantics within the given context  (via attribution and substitution scores) and the language model to evaluate the overall word likelihood given the context (via the importance score). Our context-aware method is thus capable of handling polysemy and homonymy.
>
> ---
> > **2. Question**: In SDLG, when a sentence (sentence B) is generated during the uncertainty measurement of a sentence (sentence A) after substitution, is it possible to reuse the semantic and uncertainty-related information obtained to directly measure the uncertainty of sentence B, or must a new sampling and evaluation process be initiated?
>
> In general, predictive uncertainty estimation is concerned with quantifying a model's uncertainty for a given input, rather than for specific outputs it generates. In our work, we adhere to this principle by measuring the uncertainty of the language model when prompted with a particular input sequence (e.g. a trivia question), rather than evaluating the uncertainty of individually generated output sequences (e.g., sentences A and B).
>
> ---
> > **3. Question**: How would different NLI models, such as smaller versions of DeBERTa, influence the results? Can the robustness of SDLG be maintained across different NLI models? (simultaneously answering the 1. point of your stated weaknesses)
>
> We agree that biases inherent in the NLI model could influence uncertainty estimates. However, it is important to emphasize that any potential biases in the NLI model would affect all methods that rely on it, including the baselines we compare against [3,4]. Thus, further investigations into NLI models would be an interesting research direction on its own but would go beyond the scope of this work.
> To ensure a fair comparison, we isolate the impact of our improved sampling strategy by using the same NLI model as Kuhn et al. [3]. The results clearly demonstrate that our method achieves superior performance under these controlled conditions, indicating that the improvement stems from the sampling procedure itself and not the choice of a specific NLI model.
> However, to address your recommendation, we are currently running the experiments with a smaller NLI model. We will include this ablation study in the camera-ready version of the paper, as the experiments, unfortunately, take too long for results to be available within the rebuttal period.
>
> ---
> > **4. Question**: In cases where there are no existing NLI models with an identical tokenizer, can you fine-tune a small NLI model from scratch for a novel LLM with any vocabulary? (simultaneously answering the 2. point of your stated weaknesses)
>
> This is a thoughtful observation, and we appreciate your attention to detail. As briefly described in lines 934 to 939 in the Appendix, our method can be applied even when the NLI model does not share the same tokenizer as the language model. Artetxe et al. [5] demonstrate that a differentiable bilingual mapping can align token embeddings into a shared space, enabling our method to be used without any modifications. Alternatively, a dedicated NLI model with the same tokenizer as the language model could of course also be created, either by fine-tuning an existing model or training one from scratch. However, as mentioned above, we have not explored these approaches further, since investigations into the NLI model fall outside the scope of this work and our primary focus is on isolating the impact of our improved sampling strategy within the current setting.

---

> > ### Author Response · Authors · 2024-11-21
> >
> > > **5. Question**: How is the correctness of uncertainty estimation evaluated, and what are the implications for downstream tasks?
> >
> > As stated in lines 403 to 414, we follow prior work by evaluating the performance of an uncertainty estimator based on its correlation with the correctness of the generated answer. To assess the correctness of the generated answer, we again follow prior work and use the statistics-based metrics Rouge-L and Rouge-1. They consider the longest common subsequence and the overlap of unigrams between the generated answer and the ground truth answer, respectively. Additionally, we use the transfer learning-based metric BLEURT, which indicates how well the generated answer conveys the meaning of the ground truth answer. To ensure a comprehensive evaluation, we apply ten correctness thresholds ranging from 0.1 to 1.0 (exact match), resulting in a total of 30 evaluation settings for each answer. This extensive setup demonstrates that our method reliably measures the uncertainty of answers, making it applicable to a variety of downstream tasks. We have revised the paper to make this more clear.
> >
> > ---
> > > **6. Question**: Could you provide more explanations for the figures and tables in the appendix, specifically figure 9 and table 3?
> >
> > Thank you for pointing this out, we added a more detailed explanation for these figures and tables in the updated version of our paper to ensure clarity and provide additional context.
> >
> > ---
> > > **7. Question**: Could you explain why, in Table 3, multinomial sampling approaches tend to generate completely identical outputs?
> >
> > Since multinomial sampling does not consider the semantics of the generated output sequence but only the likelihood on a token level, it often generates identical answers. To give an illustrative example, if the answer “Yes” has a likelihood of 90% and the answer “No” has a likelihood of 10%, using multinomial sampling to generate ten answers would, on expectation, result in nine times “Yes” and “No” only once. In contrast, SDLG would generate “No” already at the second generation and account for the lower likelihood by a lower importance weight for this second generation.
> >
> > ---
> > > **8. Question**: How does SDLG compare to approaches that evaluate self-consistency with beam-search sampling like [1,2] in terms of assessing NLG uncertainty?
> >
> > Thank you for this excellent question. Methods like [1, 2] rely on generating diverse candidate reasoning paths, with these two methods explicitly proposing multinomial sampling or beam search. These naïve sampling methods are also commonly used for sampling different output sequences for uncertainty estimation in NLG. Since SDLG demonstrates superior semantic diversity in its output sequences, it could potentially enhance the field of reasoning and decision-making as well.  This is a promising avenue for future work.
> >
> > ---
> > > **3. Weakness**: Due to space constraints, the current main paper does not include a detailed illustration of its scoring mechanism and how the specific token pairs are selected and replaced. I recommend reducing sections 2 and 7 or relocating parts of them to the appendix.
> >
> > Could you clarify which specific aspects you would like to see elaborated further? We have already provided a detailed verbal explanation of both the scoring mechanism and token selection process but are happy to further illustrate any specific part in case you see a need for it. Your feedback will help us address your concerns more effectively.
> >
> > ---
> > Overall, we hope that our clarifications and revisions address your concerns and further strengthen your view of our work. Should you have any further inquiries, we look forward to addressing them. Otherwise, we hope for a positive reassessment of our work.
> >
> > ---
> > [1] X. Wang, J. Wei, D. Schuurmans, Q. Le, E. Chi, S. Narang, A. Chowdhery, and D. Zhou, "Self-consistency improves chain of thought reasoning in language models," *arXiv preprint arXiv:2203.11171*, 2022.
> >
> > [2] Y. Xie, K. Kawaguchi, Y. Zhao, X. Zhao, M. Kan, J. He, and Q. Xie, "Self-evaluation guided beam search for reasoning," *Advances in Neural Information Processing Systems*, vol. 36, 2024.
> >
> > [3] L. Kuhn, Y. Gal, and S. Farquhar, "Semantic uncertainty: Linguistic invariances for uncertainty estimation in natural language generation," *arXiv preprint arXiv:2302.09664*, 2023.
> >
> > [4] J. Duan, H. Cheng, S. Wang, A. Zavalny, C. Wang, R. Xu, B. Kailkhura, and K. Xu, "Shifting attention to relevance: Towards the predictive uncertainty quantification of free-form large language models," *arXiv preprint arXiv:2307.01379*, 2023.
> >
> > [5] M. Artetxe, G. Labaka, and E. Agirre, "Learning principled bilingual mappings of word embeddings while preserving monolingual invariance," in *Proc. Conf. Empirical Methods in Natural Language Processing (EMNLP)*, 2016.

---

> ### Comment · Reviewer_KtLw · 2024-11-26
>
> Thanks for the clarifications, which have addressed most of my concerns. For Q3, do you have any preliminary experimental results regarding smaller NLI models? For W3, I recommend relocating a condensed version of Figure 11 to the main paper, ideally accompanied by an example, to illustrate the scoring mechanism more clearly.

---

> > ### Author Response · Authors · 2024-11-26
> >
> > Thank you for your valuable feedback. We are glad that our rebuttal has addressed most of your concerns!
> >
> > ---
> > **Q3**: Preliminary experiments with the smaller NLI model indicate that the performance of the uncertainty methods remains comparable, with our method continuing to outperform the baselines on the three considered datasets.
> >
> > ---
> > **W3**: We appreciate your suggestion and will make every effort to relocate a condensed version of Figure 11 to the main paper.
> >
> > ---
> > We hope these updates address your remaining concerns and further demonstrate the effectiveness of our method.

---

> > > ### Author Response · Authors · 2024-12-01
> > >
> > > With the rebuttal phase coming to a close, we kindly follow up to check whether our responses have addressed your remaining questions. If so, we would be grateful for this to be reflected in your final assessment of our work.  If any questions persist, we would be happy to discuss them during the final day of the rebuttal phase.

---

### Author Response · Authors · 2024-11-25

Dear Reviewers,

As the end of the rebuttal phase is approaching, we kindly invite you to take a look at our responses and the revised version of our paper.  We have worked diligently to address all questions and concerns raised and to substantially improve the paper based on your valuable feedback. If you have any additional questions or require further clarification, we would be more than happy to address them.

Thank you once again for your insightful comments and the effort you have dedicated to reviewing our work.

---

### Meta-Review · Area_Chair_DAEr · 2024-12-23

**Metareview:**

The paper presents a method called SLDG to enhance uncertainty measurement in language generation models.  The novelty lies in using semantic divergence of the generated sequences to measure uncertainty.  The method results in better uncertainty estimation than SOTA.  Other than utilizing NLI models that can have noise (as pointed out by reviewers), I found the paper to be quite strong:  with good writing, well motivated, good experimental design.  This is why I think the paper should be accepted.

**Additional Comments On Reviewer Discussion:**

There has been a good set of discussions between authors and reviewers for this paper which resulted in an improvement mutual understanding of the paper.

---

### Decision · Program_Chairs · 2025-01-22

Accept (Poster)